# SELF-SUPERVISED LEARNING TO PREDICT OPTIMIZER PERFORMANCE: A TRADE-OFF STUDY BETWEEN GROUND TRUTH REQUIREMENTS AND PREDICTION QUALITY

## ABSTRACT

Predicting optimizer performance typically requires costly ground-truth performance data, since each label requires multiple executions of optimization algorithms, which are computationally expensive to obtain on a large set of problems. We investigate whether established self-supervised learning (SSL) can reduce this labeling burden while preserving performance. Using four landscape representations (ELA, DeepELA, DoE2Vec, TransOptAS), seven learners (Random Forest (RF), XGBoost, LightGBM, SCARF, TabNet, VIME, and SAINT), and four train–test splitting strategies with increasing distribution shift (I, R, PC, P), we quantify the label–quality trade-off for predicting the performance of five Differential Evolution (DE) and five Particle Swarm Optimization (PSO) configurations in a controlled comparative study. We find that across settings, SSL, especially SCARF on learned representations, matches or surpasses supervised RF with only 50–75% labels. Under moderate shift (PC), SSL yields substantial benchmarking gains: for multiple configurations, we can omit around 80% of function evaluations without worsening MAE, with improvements visible already at 25% labels. Computed ELA remains a strong supervised baseline in low-shift regimes (I/R), whereas all representations amplify SSL benefits under shift (PC). The zero-shot split (P) remains challenging, motivating richer problem-space augmentation. The results confirm that SSL benefits are portfolio-agnostic (across DE and PSO). A practical guideline emerges: use RF+ELA for in-distribution tasks, while when facing distribution shifts, apply Doe2Vec with SCARF pre-training and fine-tuning on 50–75% of labels, which achieves performance close to fully supervised models while reducing the need for costly function evaluations.

## 1 INTRODUCTION

Black-box optimization (BBO) plays a critical role in many areas of AI and ML, especially in settings where objective functions are unknown, non-differentiable, or expensive to evaluate. It relies on sampling and evaluating candidate solutions without access to gradients or internal structure. While metaphor-driven metaheuristics have been widely used in BBO, the evolutionary computation community has raised concerns over their scientific rigor, lack of novelty (Sörensen, 2015), and biased benchmarking practices (Bartz-Beielstein et al., 2020). These issues have led to algorithm proliferation and hindered a deeper understanding of algorithm behavior across diverse problem landscapes. A central open challenge is learning to select the most appropriate optimization algorithm (or configuration) for a given problem. This task, known as automated algorithm selection (AAS) (Kerschke et al., 2019; Alissa et al., 2023), has increasingly relied on meta-learning approaches that map problem representations to expected algorithm performance. Historically, this has involved hand-crafted or computed meta-features describing the problem landscape, while more recently, deep learning models that learn such representations directly from data have been explored. However, supervised AAS approaches demand extensive ground-truth performance data, which is computationally expensive to obtain, since it involves the execution of several algorithms across all problems in the training data, with multiple repetitions performed with different random seeds

to account for stochasticity and obtain reliable performance estimates. Given the stochasticity of optimization algorithms and the diversity of problem landscapes, collecting this data at scale is resource-intensive and often impractical.

To address these challenges, recent work in single-objective continuous optimization (SOO) has explored representation learning for problem landscapes. Techniques include Exploratory Landscape Analysis (ELA) (Mersmann et al., 2011), topological approaches (Petelin et al., 2024), and deep architectures such as autoencoders (e.g., DoE2Vec) (van Stein et al., 2023), convolutional neural networks (CNNs) (e.g., FitnessMap) (Prager et al., 2021; Seiler et al., 2022), and transformers (e.g., DeepELA (Seiler et al., 2025), TransOptAS (Cenikj et al., 2024)). These methods aim to encode optimization problems into fixed-size representations that can be used to predict algorithm performance. Yet, despite their promise, prediction performance tends to degrade under distribution shifts between train and test data, especially in zero-shot scenarios (Petelin & Cenikj, 2023; Cenikj et al., 2025). Moreover, the choice of benchmark dataset significantly influences results, raising concerns about generalization (Petelin & Cenikj, 2023; Tanabe, 2022).

In this paper, we empirically investigate how much ground-truth performance data is required to reliably predict optimization performance, focusing on reducing the cost of benchmarking through more efficient learning. We model the trade-off between data availability and prediction quality by varying the proportion of problem instances with known performance and applying self-supervised learning (SSL) to infer missing labels. This allows us to estimate how many function evaluations can be saved while still achieving reliable prediction outcomes. Our findings contribute to the emerging paradigm of green benchmarking, promoting more sustainable and scalable evaluation practices. We conduct our study on two widely-used algorithm families: Differential Evolution (DE) (Pant et al., 2020) and Particle Swarm Optimization (PSO) (Wang et al., 2018). For each, we predict the performance of five hyperparameter configurations using four state-of-the-art landscape representations—ELA, DeepELA, DoE2Vec, and TransOptAS. We assess generalization performance under four levels of train/test distribution shift, including a zero-shot setup. The results reveal consistent performance patterns and offer practical insights into which representations are most data-efficient, helping guide future AAS approaches in resource-constrained environments.

**Outline:** The rest of the paper is organized as follows: Section 2 presents the related work. Section 3 describes the methodology used in our analysis. Experimental settings are detailed in Section 4, followed by results in Section 5. Finally, Section 6 provides a discussion of the results, concludes the paper, and outlines directions for future work.

## 2 RELATED WORK

We begin by outlining the background of continuous single-objective optimization (SOO), followed by an overview of learning-based approaches for automated algorithm performance prediction, and conclude with recent advances in label-efficient learning methods that motivate our study.

**Single-objective continuous optimization:** In single-objective continuous optimization (SOO), the goal is to find the optimal solution $\mathbf{x}^* \in \mathbb{R}^n$ that minimizes a real-valued objective function $f : \mathbb{R}^n \to \mathbb{R}$, i.e., $\mathbf{x}^* = \arg\min_{\mathbf{x} \in \mathbb{R}^n} f(\mathbf{x})$. A classical example is the Sphere function, $f(\mathbf{x}) = \sum_{i=1}^{n} x_i^2$, where $\mathbf{x} = (x_1, x_2, \ldots, x_n)$. Optimization algorithms iteratively generate and evaluate candidate solutions, often through stochastic sampling over the search space. Metaheuristics, such as population-based or single-solution-based approaches, are widely used due to their robustness in solving complex, noisy, or constrained problems.

**Automated algorithm selection:** AAS aims to identify the most suitable algorithm from a portfolio based on representations of the problem instance. This is typically framed as a supervised learning task, using multi-class or multi-label classification, or single- or multi-target regression (Kostovska et al., 2023). Regression is often preferred, as it can capture fine-grained performance differences between algorithms and offers robustness when the top-predicted algorithm is close in quality to the actual best. While many studies focus on AAS as a decision-making process, our focus in this paper is on the core regression task of predicting algorithm performance from problem representations (Jankovic & Doerr, 2020). Early AAS research used tabular representations based on Exploratory Landscape Analysis (ELA) (Mersmann et al., 2011) features, coupled with tree-based models such as decision trees and random forests. Explainable models, particularly those using

SHAP (Rozemberczki et al., 2022), have helped identify which features are most informative for specific problem landscapes (van Stein et al., 2025). More recently, alternative representations have been proposed, including those based on topological data analysis (e.g., TinyTLA (Petelin et al., 2024)), autoencoders (e.g., DoE2Vec (van Stein et al., 2023)), convolutional neural networks (e.g., FitnessMap (Prager et al., 2021; Seiler et al., 2022)), and transformers (e.g., DeepELA (Seiler et al., 2025; 2024), TransOptAS (Cenikj et al., 2024)). These representations have been used either in conjunction with traditional models or in end-to-end deep learning pipelines. In (Cenikj et al., 2025), a comprehensive comparison of state-of-the-art representations and ensemble of different representations was conducted using regression-based performance prediction. While results are promising, a key challenge remains: determining how much ground-truth performance data is needed to achieve reliable prediction for a given representation. Too little data may result in models that fail to generalize, while excessive labeling effort may not improve accuracy but instead increase computational costs and risk overfitting. To go beyond flat tabular representations, recent approaches structure benchmarking data as meta-graphs, where nodes represent algorithms or problems and edges capture relationships such as hyperparameter configurations, feature similarities, or observed performances. Graph neural networks (GNNs) have been used to predict algorithm performance at the node level, showing improved performance over tabular models under certain conditions, particularly depending on the optimization budget and problem dimensionality (Kostovska et al., 2025).

**Self-supervised learning:** SSL reduces reliance on task labels by learning representations from unlabeled data by creating pretext objectives, which provide supervisory signals from the data itself (Chen et al., 2020; Misra & van der Maaten, 2020). This approach has driven major advances in computer vision (Chen et al., 2020; He et al., 2020; Grill et al., 2020), and natural language processing (Devlin et al., 2019; Radford et al., 2018). In the domain of tabular data, recent work has shown that carefully designed pretext tasks, such as feature masking (Yoon et al., 2020; Arik & Pfister, 2021), latent space perturbation (Nam et al., 2023), contrastive learning (Bahri et al., 2022; Hajiramezanali et al., 2022), or a combination of these tasks (Uçar et al., 2021; Somepalli et al., 2021; Chen et al., 2023), can improve performance over purely supervised models. Consequently, SSL can reduce the need for collecting costly ground-truth performance labels, while preserving predictive power, making it well-suited for sustainable benchmarking and performance prediction in black-box optimization. In this domain, inputs are modeled as non-sequential tabular data. A recent survey (Wang et al., 2025) systematically categorizes self-supervised methods for non-sequential tabular data into three paradigms - predictive (masking/reconstruction), contrastive (view-based similarity), and hybrid approaches.

## 3 METHODOLOGY

We empirically investigate the trade-off between the amount of ground-truth performance data required and the quality of algorithm performance prediction. Specifically, we aim to quantify how much labeled data is necessary to build accurate predictive models across different problem representations and under varying evaluation conditions. Let $\mathcal{P} = \{p_1, p_2, \ldots, p_N\}$ denote a set of optimization problem instances, each represented by $\phi(p_i) \in \mathbb{R}^d$, where $\phi$ is a problem representation method (e.g., ELA, DeepELA, DoE2Vec, or TransOptAS). Each instance $p_i$ is associated with a ground-truth performance value $y_i \in \mathbb{R}$ for a given algorithm configuration, measured under a fixed optimization budget. The prediction task is formulated as learning a function $f : \mathbb{R}^d \to \mathbb{R}$ such that $f(\phi(p_i)) \approx y_i$. To evaluate generalization behavior, we consider four evaluation scenarios with increasing distributional shift between training and test data (see Section 4). To simulate limited supervision, we vary the proportion of labeled training data $\mathcal{L} \subset \mathcal{P}$ used to train $f$, from 25%, 50%, 75%, to 100%, leaving $\mathcal{U} = \mathcal{P} \setminus \mathcal{L}$ as the unlabeled portion of the data. For the SSL models, we use all available data in the pre-training phase, and the labeled portion $\mathcal{L}$ for fine-tuning phase. This setting allows us to evaluate label efficiency across different representations and conditions. Our analysis addresses the following questions: (i) Assess whether a given representation $\phi$ can support accurate performance prediction under limited supervision across various train–test distribution shifts; and (ii) Quantify the number of function evaluations that can be omitted by reducing the labeling budget without significantly degrading model performance. This methodology allows us to assess the label-efficiency, generalization, and sustainability of learning-based performance predictors in black-box single-objective continuous optimization.

## 4 EXPERIMENTAL DESIGN

The data used in this study is sourced from (Cenikj et al., 2025); a brief summary is provided below.
**Problem benchmark data:** The first five instances ($m, n \in \{1, \ldots, 5\}$) of each of the 24 Black-Box Optimization Benchmarking (BBOB) (Hansen et al., 2009) problem classes have been used to generate affine recombinations, following the method proposed by (Vermetten et al., 2025). Each BBOB problem class defines a function type (e.g., sphere, Rosenbrock), while different instances within the same class share the same functional form but differ in transformations such as shifts, rotations, or scalings. These variations preserve problem structure while introducing diversity to test algorithm robustness. Two problem instances $P_{i,m}$ and $P_{j,n}$ (from different classes, $i \neq j$, with matching instance indices $m = n$) are blended using an affine transformation controlled by a parameter $\alpha \in \{0.25, 0.5, 0.75\}$, according to the following formulation: $F(P_{i,m}, P_{j,n}, \alpha)(x) = \exp\left(\alpha \log(P_{i,m}(x) - P_{i,m}(O_{i,m})) + (1-\alpha) \log(P_{j,n}(x - O_{i,m} + O_{j,n}) - P_{j,n}(O_{j,n}))\right)$, where $O_{a,b}$ denotes the location of the optimum of function $P_{a,b}$. Each instance from one problem class is recombined with the corresponding instance from the other 23 classes, producing a total of 8,280 affine-transformed problem instances. To evaluate each instance, we generate a Latin Hypercube Sample (Menčík, 2016) of size $50d$ (where $d$ is the problem dimension) and compute the objective values over the sampled points. We fixed the problem dimension at 10 ($d = 10$).

**Problem representations:** Four state-of-the-art problem representations were evaluated: ELA, Doe2Vec, TransOptAS, and DeepELA. ELA features are computed using the `flacco` Python library (Prager & Trautmann, 2024), selecting only feature groups that do not require additional function evaluations, resulting in 62 features per problem instance. Doe2Vec generates a 32-dimensional latent representation using a variational autoencoder trained to reconstruct the objective function values of 250,000 randomly generated problems (van Stein et al., 2023). A fixed sample of candidate solutions is used for both training and inference to ensure comparability across all representations. TransOptAS learns 50-dimensional embeddings through a transformer encoder trained to predict algorithm performance on 30,000 synthetic problems (Cenikj et al., 2024). Once trained, the regression layer is removed and the next-to-last layer is used to extract features, with separate models trained for each algorithm portfolio (in our case, the PSO portfolio). DeepELA uses a transformer model trained in a self-supervised way to provide representations that are not sensitive to function transformations (Seiler et al., 2025). We evaluate the pre-trained single forward pass variant without performing any further training, generating 48 features for each problem instance. All three of the aforementioned features based on deep learning are trained on a set of functions generated using a random function generator (Tian et al., 2020), which are completely different from the data used for training the algorithm performance prediction model later on.

**Algorithm performance data:** Two algorithm portfolios were used, each consisting of different configurations of Differential Evolution (DE) (Pant et al., 2020) and Particle Swarm Optimization (PSO) (Wang et al., 2018). The full list of parameter settings follows those defined by Cenikj et al. (2025). All configurations from both algorithm portfolios were executed with a population size of $10 \times d$. To account for the stochastic nature of the algorithms, each configuration was run 10 times independently on every problem instance. A fixed-budget scenario was used to assess algorithm performance, where the best-found objective function value was recorded after a predefined number of iterations (100). Performance was measured using the normalized precision metric Cenikj et al. (2025). In total, each algorithm-problem pair was allocated 100,000 function evaluations (100 iterations $\times$ 10 $\times d$ = 10 $\times$ 10 runs).

**Data-splitting evaluation scenarios:** To capture the range of evaluation strategies used in the literature and assess the robustness of different problem representations, four evaluation settings were considered as defined by Cenikj et al. (2025). In the *Instance Split (I)*, one instance per affine problem (out of five) was assigned to the test set, with the remaining instances used for training, ensuring that the test set always included problems derived from parent classes seen during training. The *Random Split (R)* assigned instances to train and test sets without constraints, possibly resulting in overlapping or completely disjoint distributions of problem pairs across folds. In the *Problem Combination Split (PC)*, the test set consisted of instances generated by combining one fixed problem class with all others, while the training set excluded that class entirely, providing partial overlap in parent class representation. Finally, the *Problem Split (P)* setting excluded both parent classes used to generate test instances from the training set, creating a zero-shot generalization scenario where no problem class overlap existed between training and test data. Each splitting scenario has

been done five times, with different instances being used for testing in the I split, and randomly chosen problem classes being used in the P and PC splits. Following the above-mentioned scenarios, we obtain the following train and test sizes for the models: 6,624 train and 1,657 test instances for the I split; 5,796 train and 2,485 test instances for the R split; 5,700 train and 181 test instances for the P split; and 7,590 train and 346 test instances for the PC split.

**Label-efficient splits:** As the experiments involve self-supervised learning, the training data from each split has been further modified by retaining only a portion of labeled instances—specifically {25%, 50%, 75%, 100%}—while the remaining instances have been treated as unlabeled (i.e., not having the ground truth performance achieved). For each percentage, five different labeled subsets have been generated, following the logic of the corresponding stratified splitting scenario. For the I, P, and PC splits, we performed an additional evaluation in which the percentage of labeled instances in the training set was selected randomly, without preserving the original stratification. We tag these experiments with a stratification label RLS (Random Labeled Selection), RLS = FALSE (when stratification like original splits was followed) or TRUE (when the labeled instances have been randomly selected), respectively. We need to mention that when RLS = FALSE and RLS = TRUE in case of the R split is the same experiment. For the other splits (I, PC, and P), when RLS = FALSE we have a more controlled evaluation setting. The same random seeds were used across all representations and evaluation settings to ensure fair and consistent benchmarking.

**Self-supervised methods and baselines:** In this study, we have included four representative SSL approaches for tabular data, namely SCARF (Bahri et al., 2022), TabNet (Arik & Pfister, 2021), VIME (Yoon et al., 2020), and SAINT (Somepalli et al., 2021). SCARF applies contrastive learning to tabular data by generating positive views via random feature corruption. TabNet uses sequential attention to select a subset of features to use at each decision step, enabling interpretability and efficient use of representational capacity. TabNet is pre-trained with a reconstruction objective (masked column prediction) and then fine-tuned. VIME's self-supervised learning component learns feature representations by masking input features and training the model to reconstruct both the original values and predict which features were masked. SAINT uses inter-sample attention to learn relationships between samples and combines contrastive with denoising objectives to learn robust representations before fine-tuning the encoder on the labeled data.

We include Random Forest (Breiman, 2001) (RF), XGBoost (Chen & Guestrin, 2016), and LightGBM (Ke et al., 2017) regressors as supervised baselines, and a naive mean predictor that always predicts the training mean of each target. Baselines are trained only on the specified fraction of labeled training instances. Results achieved by SCARF and TabNet are discussed in detail and compared against the best-performing supervised baseline (RF) as well as the naive mean predictor. The remaining results achieved by VIME, SAINT, XGBoost, and LightGBM are provided in Appendix D. While they do not alter the main findings, they offer additional context and further reinforce the conclusions of this study. We frame performance prediction as multi-target regression with five outputs, each corresponding to a distinct configuration in the optimizer portfolio. All features are z-normalized (mean subtraction and unit variance).

## 5 RESULTS

We report selected key findings, with the full experiments available in our repository. Results are shown for the DE portfolio (PSO yields consistent patterns online). We analyze performance across four feature representations (DeepELA, Doe2Vec, ELA, TransOptAS), all evaluation splits (I, R, PC, P), and both sampling strategies (RLS = FALSE, RLS = TRUE). Each representation is paired with either a supervised learner (RF) or a self-supervised learner (SCARF, TabNet). We conducted ablation studies on multiple hyperparameters for both SCARF and TabNet. The results show that SCARF remains consistently stable, while TabNet exhibits only minor variation. Importantly, these variations do not affect the overall outcomes or conclusions of the study (see Appendix C). Finally, we quantify efficiency gains of self-supervised learning by reporting the percentage of function evaluations omitted relative to the RF baseline trained with 100% labeled data, across all settings.

**Results by Feature Representation and Data Splits:** Figure 1 reports the mean absolute error (MAE) averaged over five DE configurations, shown separately for the four feature representations across evaluation splits (I, R, PC, P) under stratified label selection (RLS = FALSE). Results for mean square error (MSE) and root mean square error (RMSE) are following similar trends, and

are available in Appendix A. Columns correspond to evaluation splits and rows to feature representations (DeepELA, Doe2Vec, ELA, TransOptAS). Each subplot compares a naive baseline (mean target of labeled training instances), a supervised RF model trained with the portion of labeled instances, and two self-supervised models (SCARF and TabNet). The (RLS = TRUE) results are available in Figure 3 of Appendix A. **DeepELA.** In the I split, the naive baseline is worst; RF and SCARF dominate, with SCARF surpassing RF from 0.50 and lowest at 1.00, while TabNet improves but remains weaker. In the R split, we observe that RF is the best performer, with SCARF matching it at 0.75. In the PC split, errors rise; SCARF is consistently best across all ratios, while RF remains competitive and the naive baseline/TabNet lag. In the P split, differences shrink, with TabNet unstable at 0.25 but later converging. **Doe2Vec.** In the I split, RF remains the best performing model across all labeled scenarios, with SCARF tailing it closely; naive baseline and TabNet remain weak. The R split follows: RF is slightly better at 0.25, but SCARF is the best performer from 0.50 onward. In the PC split, RF performs the best in 0.25 and 0.5 scenarios, SCARF takes over from 0.75 on, while naive baseline and TabNet underperform. In the P split, all converge, with TabNet unstable at 0.25. **ELA.** Across I and R splits, naive baseline and TabNet perform the worst, while RF consistently outperforms SCARF across ratios. In the PC split, RF and SCARF dominate, with SCARF trending slightly lower. In the P split, RF and the naive baseline are stable with low errors; SCARF fluctuates; TabNet is unstable at 0.25. **TransOptAS.** In the I split, SCARF is lowest in the case of 0.75 and 1.00 ratios, RF tailing closely, and TabNet improving to near RF at 1.00. The R split is similar: SCARF is the best from 0.5 on, RF is the best at 0.25 and is slightly above after 0.5, TabNet never closes the gap. In the PC split, SCARF maintains the lowest errors, RF/TabNet just above, and the naive baseline is the worst. In the P split, the naive baseline is consistently lowest, RF and SCARF are similar with gradual gains, while TabNet is unstable at 0.25 due to a large default batch size. A smaller batch size stabilized TabNet in this setting, as shown in Figure 13 in Appendix C, but remains inferior to the supervised baseline.

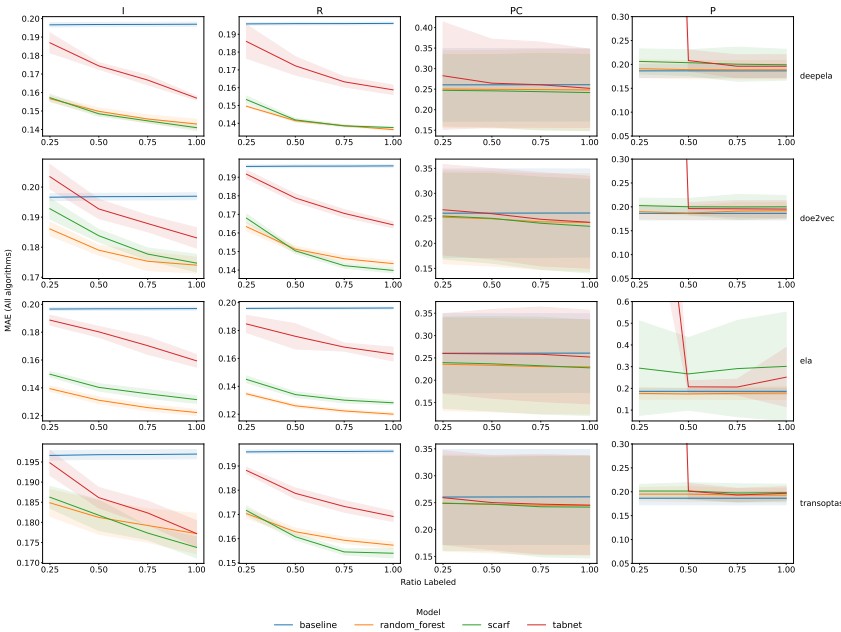

Figure 1: Mean Absolute Error (MAE) averaged over all five DE configurations, presented separately for each of the four feature representations, across the data–evaluation splits (I, R, PC, and P) using the stratified selection of labeled instances with respect to the data–evaluation split RLS = FALSE.

**Results by Learning Model and Data Splits:** Table 1 presents the MAE (mean ± sd) aggregated across all DE configurations for each feature representation and data–evaluation split, with/without stratified labeled instance selection, and for different ratios of labeled instances. Table 3 from the Appendix F presents the same results for the PSO algorithm portfolio. We observe a consistent pattern regardless of how the labeled instances are selected (RLS = TRUE or FALSE). In the I split,

Table 1: Mean Absolute Error (MAE; mean ± sd) across feature representations and data–evaluation splits with/without stratified labeled instance selection (RLS). The cases where SSL (either TabNet or SCARF) outperforms the baselines are bolded.

| | | RLS | False | | | | True | | | |
|---|---|---|---|---|---|---|---|---|---|---|
| | | Model | Baseline | RF | SCARF | TabNet | Baseline | RF | SCARF | TabNet |
| Evaluation split | Feature representation | Ratio Labeled | | | | | | | | |
| I | deepela | 0.25 | 0.197 ± 0.001 | **0.157 ± 0.002** | **0.157 ± 0.002** | 0.187 ± 0.005 | 0.197 ± 0.001 | 0.151 ± 0.002 | 0.154 ± 0.002 | 0.182 ± 0.001 |
| | | 0.50 | 0.197 ± 0.001 | 0.150 ± 0.002 | **0.148 ± 0.001** | 0.174 ± 0.002 | 0.197 ± 0.001 | 0.147 ± 0.002 | **0.146 ± 0.002** | 0.172 ± 0.004 |
| | | 0.75 | 0.197 ± 0.001 | 0.146 ± 0.002 | **0.145 ± 0.001** | 0.167 ± 0.003 | 0.197 ± 0.001 | 0.144 ± 0.002 | **0.143 ± 0.002** | 0.164 ± 0.001 |
| | | 1.00 | 0.197 ± 0.001 | 0.143 ± 0.003 | **0.141 ± 0.002** | 0.157 ± 0.001 | 0.197 ± 0.001 | 0.143 ± 0.003 | **0.141 ± 0.002** | 0.159 ± 0.004 |
| | doe2vec | 0.25 | 0.197 ± 0.001 | 0.186 ± 0.002 | 0.193 ± 0.003 | 0.203 ± 0.004 | 0.197 ± 0.001 | 0.177 ± 0.002 | 0.184 ± 0.003 | 0.195 ± 0.002 |
| | | 0.50 | 0.197 ± 0.001 | 0.179 ± 0.002 | 0.184 ± 0.002 | 0.193 ± 0.003 | 0.197 ± 0.001 | 0.175 ± 0.002 | 0.179 ± 0.002 | 0.190 ± 0.003 |
| | | 0.75 | 0.197 ± 0.001 | 0.175 ± 0.003 | 0.178 ± 0.002 | 0.188 ± 0.003 | 0.197 ± 0.001 | **0.175 ± 0.003** | **0.175 ± 0.003** | 0.187 ± 0.002 |
| | | 1.00 | 0.197 ± 0.001 | 0.174 ± 0.003 | 0.175 ± 0.003 | 0.183 ± 0.003 | 0.197 ± 0.001 | 0.174 ± 0.003 | 0.175 ± 0.003 | 0.184 ± 0.004 |
| | ela | 0.25 | 0.197 ± 0.001 | 0.139 ± 0.002 | 0.150 ± 0.002 | 0.189 ± 0.004 | 0.197 ± 0.001 | 0.136 ± 0.002 | 0.145 ± 0.002 | 0.202 ± 0.021 |
| | | 0.50 | 0.197 ± 0.001 | 0.131 ± 0.002 | 0.140 ± 0.003 | 0.180 ± 0.004 | 0.197 ± 0.001 | 0.129 ± 0.002 | 0.138 ± 0.002 | 0.177 ± 0.003 |
| | | 0.75 | 0.197 ± 0.001 | 0.126 ± 0.002 | 0.136 ± 0.003 | 0.17 ± 0.006 | 0.197 ± 0.001 | 0.125 ± 0.002 | 0.134 ± 0.002 | 0.169 ± 0.002 |
| | | 1.00 | 0.197 ± 0.001 | 0.122 ± 0.002 | 0.132 ± 0.003 | 0.159 ± 0.005 | 0.197 ± 0.001 | 0.122 ± 0.002 | 0.131 ± 0.002 | 0.159 ± 0.007 |
| | transoptas | 0.25 | 0.197 ± 0.001 | 0.185 ± 0.003 | 0.186 ± 0.003 | 0.195 ± 0.003 | 0.197 ± 0.001 | 0.179 ± 0.004 | **0.178 ± 0.005** | 0.189 ± 0.003 |
| | | 0.50 | 0.197 ± 0.001 | 0.181 ± 0.004 | 0.182 ± 0.004 | 0.186 ± 0.003 | 0.197 ± 0.001 | 0.178 ± 0.004 | **0.177 ± 0.004** | 0.185 ± 0.003 |
| | | 0.750 | 0.197 ± 0.001 | 0.179 ± 0.004 | **0.177 ± 0.002** | 0.182 ± 0.003 | 0.197 ± 0.001 | 0.178 ± 0.005 | **0.176 ± 0.003** | 0.182 ± 0.004 |
| | | 1.0 | 0.197 ± 0.001 | 0.177 ± 0.005 | **0.174 ± 0.003** | 0.177 ± 0.003 | 0.197 ± 0.001 | 0.177 ± 0.005 | **0.174 ± 0.004** | **0.177 ± 0.004** |
| R | deepela | 0.25 | 0.196 ± 0.001 | 0.150 ± 0.000 | 0.153 ± 0.002 | 0.186 ± 0.009 | / | / | / | / |
| | | 0.50 | 0.196 ± 0.001 | 0.141 ± 0.001 | 0.142 ± 0.001 | 0.172 ± 0.005 | / | / | / | / |
| | | 0.75 | 0.196 ± 0.001 | **0.139 ± 0.000** | **0.139 ± 0.001** | 0.163 ± 0.003 | / | / | / | / |
| | | 1.00 | 0.196 ± 0.001 | 0.136 ± 0.001 | 0.138 ± 0.001 | 0.159 ± 0.003 | / | / | / | / |
| | doe2vec | 0.25 | 0.196 ± 0.001 | 0.163 ± 0.002 | 0.168 ± 0.002 | 0.192 ± 0.002 | / | / | / | / |
| | | 0.50 | 0.196 ± 0.001 | 0.151 ± 0.002 | **0.150 ± 0.001** | 0.179 ± 0.002 | / | / | / | / |
| | | 0.75 | 0.196 ± 0.001 | 0.146 ± 0.002 | **0.142 ± 0.001** | 0.171 ± 0.002 | / | / | / | / |
| | | 1.00 | 0.196 ± 0.001 | 0.143 ± 0.001 | **0.140 ± 0.001** | 0.164 ± 0.002 | / | / | / | / |
| | ela | 0.25 | 0.196 ± 0.001 | 0.135 ± 0.001 | 0.145 ± 0.002 | 0.185 ± 0.006 | / | / | / | / |
| | | 0.50 | 0.196 ± 0.001 | 0.126 ± 0.001 | 0.134 ± 0.002 | 0.176 ± 0.009 | / | / | / | / |
| | | 0.75 | 0.196 ± 0.001 | 0.122 ± 0.001 | 0.130 ± 0.002 | 0.168 ± 0.003 | / | / | / | / |
| | | 1.00 | 0.196 ± 0.001 | 0.120 ± 0.001 | 0.128 ± 0.001 | 0.163 ± 0.005 | / | / | / | / |
| | transoptas | 0.25 | 0.196 ± 0.001 | 0.170 ± 0.002 | 0.172 ± 0.002 | 0.188 ± 0.001 | / | / | / | / |
| | | 0.50 | 0.196 ± 0.001 | 0.163 ± 0.002 | **0.161 ± 0.001** | 0.179 ± 0.002 | / | / | / | / |
| | | 0.75 | 0.196 ± 0.001 | 0.159 ± 0.001 | **0.155 ± 0.001** | 0.173 ± 0.002 | / | / | / | / |
| | | 1.00 | 0.196 ± 0.001 | 0.157 ± 0.001 | **0.154 ± 0.002** | 0.169 ± 0.002 | / | / | / | / |
| PC | deepela | 0.25 | 0.260 ± 0.088 | 0.250 ± 0.093 | **0.247 ± 0.087** | 0.283 ± 0.131 | 0.261 ± 0.088 | 0.250 ± 0.095 | **0.247 ± 0.092** | 0.283 ± 0.129 |
| | | 0.50 | 0.261 ± 0.088 | 0.249 ± 0.095 | **0.246 ± 0.088** | 0.265 ± 0.107 | 0.261 ± 0.088 | 0.249 ± 0.097 | **0.244 ± 0.093** | 0.259 ± 0.095 |
| | | 0.75 | 0.261 ± 0.088 | 0.249 ± 0.097 | **0.244 ± 0.093** | 0.261 ± 0.104 | 0.261 ± 0.088 | 0.249 ± 0.098 | **0.243 ± 0.091** | 0.254 ± 0.097 |
| | | 1.00 | 0.261 ± 0.088 | 0.248 ± 0.099 | **0.242 ± 0.093** | 0.252 ± 0.095 | 0.261 ± 0.088 | 0.248 ± 0.098 | **0.243 ± 0.093** | 0.254 ± 0.098 |
| | doe2vec | 0.25 | 0.260 ± 0.088 | 0.252 ± 0.093 | 0.255 ± 0.086 | 0.267 ± 0.09 | 0.261 ± 0.089 | 0.243 ± 0.097 | **0.239 ± 0.085** | 0.265 ± 0.081 |
| | | 0.50 | 0.261 ± 0.088 | 0.249 ± 0.094 | 0.250 ± 0.09 | 0.259 ± 0.09 | 0.261 ± 0.088 | 0.241 ± 0.098 | **0.235 ± 0.09** | 0.253 ± 0.088 |
| | | 0.75 | 0.261 ± 0.088 | 0.243 ± 0.096 | **0.240 ± 0.092** | 0.248 ± 0.093 | 0.261 ± 0.088 | 0.241 ± 0.098 | **0.234 ± 0.094** | 0.241 ± 0.088 |
| | | 1.00 | 0.261 ± 0.088 | 0.242 ± 0.098 | **0.234 ± 0.094** | 0.242 ± 0.092 | 0.261 ± 0.088 | 0.241 ± 0.097 | **0.236 ± 0.094** | 0.241 ± 0.093 |
| | ela | 0.25 | 0.26 ± 0.088 | 0.236 ± 0.104 | 0.239 ± 0.102 | 0.260 ± 0.089 | 0.261 ± 0.088 | 0.236 ± 0.105 | 0.241 ± 0.108 | 0.261 ± 0.091 |
| | | 0.50 | 0.261 ± 0.088 | 0.234 ± 0.104 | 0.237 ± 0.106 | 0.259 ± 0.099 | 0.261 ± 0.088 | 0.234 ± 0.107 | 0.239 ± 0.109 | 0.256 ± 0.094 |
| | | 0.75 | 0.261 ± 0.088 | 0.231 ± 0.105 | 0.232 ± 0.108 | 0.258 ± 0.106 | 0.261 ± 0.088 | 0.232 ± 0.107 | **0.231 ± 0.106** | 0.253 ± 0.100 |
| | | 1.00 | 0.261 ± 0.088 | 0.230 ± 0.106 | **0.228 ± 0.107** | 0.252 ± 0.105 | 0.261 ± 0.088 | 0.230 ± 0.107 | 0.231 ± 0.108 | 0.253 ± 0.106 |
| | transoptas | 0.25 | 0.26 ± 0.088 | **0.249 ± 0.089** | **0.249 ± 0.087** | 0.259 ± 0.087 | 0.261 ± 0.089 | 0.246 ± 0.092 | **0.243 ± 0.09** | 0.254 ± 0.087 |
| | | 0.50 | 0.261 ± 0.088 | **0.247 ± 0.09** | **0.247 ± 0.087** | 0.250 ± 0.087 | 0.261 ± 0.088 | 0.246 ± 0.093 | **0.243 ± 0.096** | 0.254 ± 0.092 |
| | | 0.75 | 0.261 ± 0.088 | 0.245 ± 0.092 | **0.242 ± 0.093** | 0.247 ± 0.093 | 0.261 ± 0.088 | 0.245 ± 0.093 | 0.247 ± 0.099 | 0.250 ± 0.095 |
| | | 1.00 | 0.261 ± 0.088 | 0.245 ± 0.092 | **0.242 ± 0.095** | **0.245 ± 0.092** | 0.261 ± 0.088 | **0.245 ± 0.092** | 0.241 ± 0.093 | 0.245 ± 0.089 |
| P | deepela | 0.25 | 0.186 ± 0.014 | 0.191 ± 0.019 | 0.206 ± 0.026 | 3.099 ± 0.625 | 0.186 ± 0.013 | 0.188 ± 0.019 | 0.208 ± 0.034 | 0.215 ± 0.035 |
| | | 0.50 | 0.187 ± 0.014 | 0.189 ± 0.019 | 0.204 ± 0.027 | 0.208 ± 0.022 | 0.187 ± 0.014 | 0.189 ± 0.019 | 0.203 ± 0.038 | 0.197 ± 0.023 |
| | | 0.75 | 0.186 ± 0.014 | 0.189 ± 0.019 | 0.200 ± 0.036 | 0.196 ± 0.024 | 0.186 ± 0.013 | 0.189 ± 0.019 | 0.199 ± 0.03 | 0.199 ± 0.024 |
| | | 1.00 | 0.186 ± 0.014 | 0.189 ± 0.019 | 0.199 ± 0.032 | 0.196 ± 0.024 | 0.186 ± 0.014 | 0.188 ± 0.019 | 0.201 ± 0.036 | 0.195 ± 0.022 |
| | doe2vec | 0.25 | 0.186 ± 0.014 | 0.190 ± 0.015 | 0.203 ± 0.016 | 3.144 ± 0.155 | 0.187 ± 0.014 | 0.193 ± 0.017 | 0.197 ± 0.019 | 0.195 ± 0.013 |
| | | 0.50 | 0.187 ± 0.014 | 0.187 ± 0.015 | 0.200 ± 0.017 | 0.196 ± 0.012 | 0.186 ± 0.014 | 0.192 ± 0.017 | 0.200 ± 0.025 | 0.196 ± 0.015 |
| | | 0.75 | 0.186 ± 0.014 | 0.191 ± 0.018 | 0.200 ± 0.026 | 0.196 ± 0.017 | 0.186 ± 0.014 | 0.192 ± 0.017 | 0.198 ± 0.022 | 0.196 ± 0.016 |
| | | 1.00 | 0.186 ± 0.014 | 0.192 ± 0.017 | 0.200 ± 0.024 | 0.195 ± 0.018 | 0.186 ± 0.014 | 0.192 ± 0.017 | 0.198 ± 0.026 | 0.195 ± 0.017 |
| | ela | 0.25 | 0.186 ± 0.014 | 0.177 ± 0.027 | 0.293 ± 0.217 | 2.12 ± 0.821 | 0.186 ± 0.013 | 0.176 ± 0.025 | 0.258 ± 0.140 | 0.198 ± 0.029 |
| | | 0.50 | 0.187 ± 0.014 | 0.174 ± 0.024 | 0.267 ± 0.167 | 0.207 ± 0.028 | 0.187 ± 0.014 | 0.177 ± 0.028 | 0.282 ± 0.201 | 0.194 ± 0.022 |
| | | 0.75 | 0.186 ± 0.014 | 0.177 ± 0.028 | 0.291 ± 0.222 | 0.206 ± 0.037 | 0.186 ± 0.013 | 0.177 ± 0.027 | 0.281 ± 0.200 | 0.189 ± 0.025 |
| | | 1.00 | 0.186 ± 0.014 | 0.179 ± 0.029 | 0.301 ± 0.250 | 0.252 ± 0.136 | 0.186 ± 0.014 | 0.179 ± 0.029 | 0.284 ± 0.208 | 0.196 ± 0.021 |
| | transoptas | 0.25 | 0.186 ± 0.014 | 0.195 ± 0.014 | 0.202 ± 0.014 | 3.375 ± 0.231 | 0.187 ± 0.014 | 0.193 ± 0.015 | 0.196 ± 0.016 | 0.198 ± 0.015 |
| | | 0.50 | 0.187 ± 0.014 | 0.195 ± 0.016 | 0.202 ± 0.017 | 0.201 ± 0.016 | 0.186 ± 0.014 | 0.193 ± 0.013 | 0.197 ± 0.018 | 0.197 ± 0.013 |
| | | 0.75 | 0.186 ± 0.014 | 0.193 ± 0.013 | 0.197 ± 0.019 | 0.193 ± 0.014 | 0.186 ± 0.014 | 0.193 ± 0.014 | 0.197 ± 0.019 | 0.193 ± 0.013 |
| | | 1.00 | 0.186 ± 0.014 | 0.193 ± 0.014 | 0.198 ± 0.018 | 0.197 ± 0.014 | 0.186 ± 0.014 | 0.193 ± 0.014 | 0.197 ± 0.017 | 0.194 ± 0.013 |

SCARF achieves slightly better results with DeepELA and TransOptAS (i.e., transformer-based features learned in unsupervised or supervised task) at higher amounts of labeled data, from 0.75–1.00 labeled ratios, while Doe2Vec (VAE-based features) consistently favors RF; stratification has no effect. In the R split, SCARF achieves lower MAE than RF for Doe2Vec (0.50-1.00) and TransOptAS (0.50-1.00), though gains are modest; DeepELA and ELA show little to no benefit from SSL. In the PC split, SCARF outperforms RF for DeepELA across all labeled ratios, and starting at 0.75 for TransOptAs (for which the two models perform equally at 0.25 and 0.50) and Doe2Vec. TabNet occasionally matches or exceeds SCARF at 1.00; even ELA with SCARF shows improvement at higher ratios (1.00). In the P split (zero-shot), all methods struggle, confirming that this setting is challenging for all combinations of methods and feature representations. We need to emphasize that DeepELA, TransOptAS, and Doe2Vec use deep-learned representations further transformed in pre-training, while ELA uses computed features transformed in the same phase. Performance degrades from the more balanced I and R splits to the less balanced PC and P splits, where the errors and un-

certainty are highest. Across these settings, self-supervised methods, specifically SCARF combined with learned representations, has the ability to outperform the supervised RF. Its abilities are exceptionally visible in the challenging PC scenario. The zero-shot P split remains challenging for all models. With ELA, a computed feature set based on statistical, nearest-neighbor, and information-theoretic descriptors, SCARF does not surpass RF in I and R, but it does in PC at higher amounts of labeled data, underscoring the challenge of generalizing to unseen problems.

**Impact of SSL on similarity semantics:** To understand what SSL embeddings capture beyond the original benchmark representations, we performed an additional analysis. For each test instance, we identified its most similar labeled training instance and computed their cosine similarity using both the original representations (ELA, DeepELA, DoE2Vec, TransOptAS) and the SSL-derived embeddings. We also measured the absolute difference in their true algorithm performance. This allows a direct comparison of how well similarity in each space reflects actual performance behavior. The analysis is carried out in the in-distribution setting (I) across all label ratios, using a single illustrative fold, a single seed to select the labeled portion of instances, and random splitting disabled (RLS = False). The presented results are calculated for the DE3 configuration. This setup reveals whether SSL reshapes the feature space so that similarity more accurately corresponds to true performance similarity.

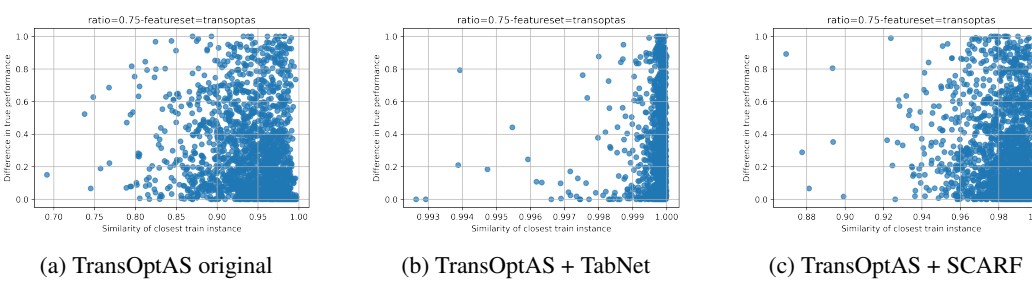

| (a) TransOptAS original | (b) TransOptAS + TabNet | (c) TransOptAS + SCARF |

Figure 2: The post-hoc analysis of whether SSL reshapes the TransOptAS feature space so that similarity more accurately corresponds to true performance similarity.

At a 75% labeled ratio (selected results where SSL shows performance gains, Figure 2), the plots show how similarity (x-axis) relates to absolute performance difference (y-axis). The x-axis shows how similar each test instance is to its closest training instance (higher is more similar), and the y-axis shows their absolute performance difference (lower is better). In a good representation, high similarity should correspond to small performance gaps—producing a tight cluster of points in the bottom-right corner. In the baseline TransOptAS space, similarities vary widely ( 0.70–1.00), and even very similar instances often show large performance gaps ($> 0.60$), indicating a noisy and weakly aligned representation. With SCARF, similarities shift upward (0.88–1.00) and form a clear cluster of very high similarity (0.98–1.00) and low performance difference (0.00–0.20), meaning the latent space becomes far more performance-aligned. TabNet, in contrast, collapses similarity into a narrow band near 1.0 (0.99–1.00), removing meaningful resolution. Large performance gaps remain even at "maximal" similarity, making similarity uninformative. TransOptAS is noisy, TabNet overcompresses similarity, while SCARF is the only method that makes similarity reliably reflect true performance differences. The analysis of ELA, DeepELA, and Doe2Vec is presented in Appendix E. Analyses for the other portions of labeled ratios are available in our repository.

**Reduction of Function Evaluations via Self-Supervised Learning:** We quantify the efficiency gains of self-supervised learning by defining the percentage of function evaluations that can be omitted when SCARF or TabNet achieves lower MAE than the supervised RF model trained with 100% of ground truth performance data. Let $\mathcal{F}$ denote the set of feature representations, $\mathcal{S}$ the data–evaluation splits, $\mathcal{L}$ the label ratios, and $\mathcal{R}$ the stratification strategies. For each configuration $(f, s, \ell, r) \in \mathcal{F} \times \mathcal{S} \times \mathcal{L} \times \mathcal{R}$, the selection of labeled instances is repeated $K = 5$ times in each of $M = 5$ folds. Within a fold $m$, we compute $p_{f,s,\ell,r}^{(m)} = \frac{1}{K} \sum_{k=1}^{K} \mathbb{1}\left[\text{MAE}_{\text{SSL}}^{(m,k)} < \text{MAE}_{\text{RF}}^{(m,k)}\right]$, where $\mathbb{1}[\cdot]$ is the indicator function that equals 1 if SSL outperforms RF with 100% labeled data in repetition $k$. We then aggregate across folds: $\bar{p}_{f,s,\ell,r} = \frac{1}{M} \sum_{m=1}^{M} p_{f,s,\ell,r}^{(m)}$. This estimator captures how often SSL provides improvements over RF across repetitions and folds. To translate this into omitted function evaluations, we scale $\bar{p}_{f,s,\ell,r}$ by the number of evaluations that would otherwise be

required in each split, label ratio, and stratification setting. We then normalize by the proportion of unlabeled instances in the scenario, so that percentages are comparable across label ratios. Finally, the percentage of evaluations that can be saved is defined as $\Delta_{f,s,\ell,r} = \bar{p}_{f,s,\ell,r} \cdot (1 - \ell) \cdot 100$, where $(1 - \ell)$ denotes the proportion of unlabeled instances. Thus, $\Delta_{f,s,\ell,r}$ measures the fraction of evaluations that can be omitted while still obtaining ground-truth performance, averaged across both repetitions and folds.

Table 2: Aggregated efficiency gains of self-supervised learning for DE, reported as the percentage of function evaluations that can be omitted ($\Delta_{f,s,\ell,r}$) when SCARF or TabNet achieves lower MAE than the supervised RF baseline trained with 100% labeled ground performance data. Results are averaged across five folds and five repetitions for each configuration of feature representation, evaluation split (I, R, PC, P), and stratification strategy (RLS = FALSE: split-aligned, RLS = TRUE: random). Label ratios of 25%, 50%, and 75% are considered, while the 100% case is omitted since all instances must be labeled. Higher values indicate greater savings in ground-truth evaluations.

| Evaluation split | | I | | | | | R | | | | | PC | | | | | P | | | | |
|---|---|---|---|---|---|---|---|---|---|---|---|---|---|---|---|---|---|---|---|---|---|
| | | | | | | | | | | | | | DeepELA | | | | | | | | |
| RLS | % labeled | DE1 | DE2 | DE3 | DE4 | DE5 | DE1 | DE2 | DE3 | DE4 | DE5 | DE1 | DE2 | DE3 | DE4 | DE5 | DE1 | DE2 | DE3 | DE4 | DE5 |
| | 0.25 | 0 | 0 | 0 | 0 | 0 | / | / | / | / | / | 40 | 40 | 80 | 40 | 60 | 20 | 60 | 0 | 0 | 40 |
| True | 0.5 | 0 | 0 | 0 | 20 | 0 | / | / | / | / | / | 40 | 40 | 100 | 60 | 60 | 40 | 40 | 40 | 40 | 60 |
| | 0.75 | 40 | 0 | 80 | 100 | 20 | / | / | / | / | / | 40 | 80 | 100 | 40 | 80 | 40 | 60 | 40 | 0 | 60 |
| | 0.25 | 0 | 0 | 0 | 0 | 0 | 0 | 0 | 0 | 0 | 0 | 21 | 43 | 85 | 43 | 64 | 0 | 0 | 0 | 0 | 43 |
| False | 0.5 | 0 | 0 | 0 | 0 | 0 | 0 | 0 | 0 | 0 | 0 | 48 | 72 | 96 | 24 | 72 | 24 | 48 | 0 | 0 | 72 |
| | 0.75 | 40 | 0 | 20 | 20 | 40 | 20 | 0 | 40 | 0 | 0 | 16 | 64 | 80 | 0 | 48 | 32 | 48 | 64 | 32 | 32 |
| | | | | | | | | | | | | | Doe2Vec | | | | | | | | |
| | 0.25 | 0 | 0 | 0 | 0 | 0 | / | / | / | / | / | 60 | 80 | 60 | 60 | 60 | 40 | 20 | 100 | 20 | 60 |
| True | 0.5 | 0 | 40 | 40 | 0 | 20 | / | / | / | / | / | 60 | 80 | 80 | 60 | 80 | 20 | 80 | 80 | 20 | 80 |
| | 0.75 | 0 | 60 | 60 | 20 | 60 | / | / | / | / | / | 80 | 80 | 100 | 40 | 80 | 40 | 80 | 60 | 20 | 80 |
| | 0.25 | 0 | 0 | 0 | 0 | 0 | 0 | 0 | 0 | 0 | 0 | 0 | 43 | 21 | 43 | 43 | 0 | 21 | 0 | 43 | 21 |
| False | 0.5 | 0 | 0 | 20 | 0 | 20 | 0 | 0 | 0 | 0 | 0 | 48 | 48 | 24 | 72 | 72 | 24 | 0 | 72 | 24 | 96 |
| | 0.75 | 0 | 40 | 40 | 0 | 60 | 40 | 0 | 100 | 20 | 20 | 48 | 48 | 48 | 16 | 64 | 16 | 80 | 48 | 16 | 32 |
| | | | | | | | | | | | | | ELA | | | | | | | | |
| | 0.25 | 0 | 0 | 0 | 0 | 0 | / | / | / | / | / | 40 | 40 | 40 | 20 | 40 | 20 | 20 | 40 | 60 | 40 |
| True | 0.5 | 0 | 0 | 0 | 0 | 0 | / | / | / | / | / | 40 | 20 | 0 | 0 | 40 | 40 | 20 | 20 | 60 | 80 |
| | 0.75 | 0 | 0 | 0 | 0 | 0 | / | / | / | / | / | 40 | 40 | 80 | 40 | 60 | 40 | 20 | 20 | 60 | 80 |
| | 0.25 | 0 | 0 | 0 | 0 | 0 | 0 | 0 | 0 | 0 | 0 | 21 | 21 | 43 | 21 | 43 | 21 | 21 | 43 | 21 | 21 |
| False | 0.5 | 0 | 0 | 0 | 0 | 0 | 0 | 0 | 0 | 0 | 0 | 48 | 72 | 96 | 48 | 48 | 0 | 24 | 48 | 24 | 72 |
| | 0.75 | 0 | 0 | 0 | 0 | 0 | 0 | 0 | 0 | 0 | 0 | 16 | 48 | 64 | 32 | 48 | 0 | 16 | 16 | 48 | 48 |
| | | | | | | | | | | | | | TransOptAS | | | | | | | | |
| | 0.25 | 40 | 20 | 60 | 20 | 60 | / | / | / | / | / | 20 | 40 | 100 | 40 | 60 | 60 | 40 | 80 | 20 | 80 |
| True | 0.5 | 40 | 60 | 60 | 20 | 40 | / | / | / | / | / | 40 | 80 | 80 | 40 | 40 | 40 | 60 | 60 | 60 | 80 |
| | 0.75 | 60 | 40 | 80 | 20 | 40 | / | / | / | / | / | 20 | 80 | 80 | 40 | 80 | 60 | 60 | 80 | 60 | 80 |
| | 0.25 | 0 | 0 | 0 | 0 | 0 | 0 | 0 | 0 | 0 | 0 | 43 | 21 | 85 | 21 | 64 | 0 | 21 | 43 | 21 | 21 |
| False | 0.5 | 20 | 40 | 20 | 0 | 0 | 0 | 0 | 0 | 0 | 0 | 48 | 72 | 100 | 48 | 48 | 48 | 24 | 24 | 24 | 72 |
| | 0.75 | 40 | 40 | 60 | 20 | 60 | 100 | 60 | 100 | 100 | 0 | 64 | 80 | 80 | 32 | 64 | 80 | 80 | 48 | 32 | 64 |

Table 2 reports the percentage of function evaluations that can be omitted when self-supervised learning (SCARF, TabNet) outperforms the supervised RF baseline trained with 100% labeled data. Unlike the previously reported aggregated results, here we present the analysis per algorithm configuration, offering more fine-grained insights. For the I and R splits, omission gains are limited or absent, reflecting the large overlap between training and test instances that reduces the benefit of including unlabeled data through SSL. In contrast, the PC split shows consistent and substantial improvements, with omission rates frequently exceeding 40–80% across multiple DE configurations (e.g., DeepELA and Doe2Vec with all five DE configurations, TransOptAS and DE3, DE5). Importantly, nontrivial gains already emerge at 25% labeled data, highlighting that SSL extracts useful structure early and scales effectively with more labels. The P split, which is a zero-shot scenario, still shows measurable omission signals: several feature–configuration pairs (e.g., DeepELA and DE2, DE5, Doe2Vec and DE1, DE3, DE5, ELA and DE3, DE4, DE5, TransOptAS DE1, DE2, DE3, DE5) achieve notable gains, even at low label ratios. Overall, learned feature representations (DeepELA, Doe2Vec, TransOptAS) amplify omission benefits relative to computed ELA, though ELA still contributes consistent improvements in the PC and P splits. While SSL provides little or no advantage in I and R, the strong and early signals in PC and P demonstrate its potential to improve robustness and generalization in distribution-shifted evaluation settings, with clear efficiency savings in ground-truth evaluations. We can also conclude that the RLS of the labeled instances favors random selection (True) over stratified sampling once (False).

We evaluate when SCARF and TabNet outperform the naive baseline, focusing on the PC and P splits (Table 4 in Appendix F). In both, SSL delivers strong omission gains, especially evident in the I and R splits. The gains are often in the range of 60–80% even at 25% labeled data. In PC, all representations quickly exceed 80% omission, and in P, several feature–algorithm configuration pairs (e.g., DeepELA DE2 (80%), Doe2Vec DE2 (60%), ELA-DE4 (60%), TransOptAS DE2–DE4 (60%)) achieve similar gains. These results show that SSL extracts generalization signals early, scales with more labels, and consistently improves robustness under distribution shift, introducing inductive biases beyond naive averaging.

## 6 DISCUSSION AND CONCLUSION

The goal of this study is to systematically assess whether existing SSL approaches, representing different SSL paradigms (TabNet as predictive and SCARF as contrastive), can reduce labeling requirements for optimizer performance prediction in a manner that is comparable across landscape representations. It demonstrates a clear label-efficiency sweet spot: self-supervised learning (SSL) achieves reliable optimizer performance prediction with only 50–75% of labeled data, reducing ground-truth requirements while preserving or even improving predictive accuracy. This highlights SSL as a viable tool for greener and more sustainable benchmarking. Performance depends strongly on the degree of train–test distribution shift. In the I and R splits (low shift), RF with computed ELA features remains strongest, although SCARF with DeepELA/TransOptAS reaches parity once 50-75% of labels are available. In the PC split (moderate shift), SCARF combined with learned representations, specifically DeepELA and TransOptAS, consistently outperforms RF, with improvements already visible at 25% labels; TabNet sometimes closes the gap at higher label ratios. The P split (zero-shot) remains the most challenging: all models degrade, yet SSL still extracts useful signals for certain representation–configuration pairs. In the PC split, SSL achieves omission rates of 40–100% across multiple DE configurations, with substantial gains already visible at 25% labeled data. In the P split, several feature–algorithm combinations also reach 60–100% omission, even under zero-shot distribution shift. We further show that representation choice is critical: learned features (DeepELA, DoE2Vec, TransOptAS) can amplify SSL gains, while computed ELA serve as a strong supervised baseline in in-distribution splits, but benefit from SSL primarily under shift (PC). Results are also robust to sampling strategy (RLS = TRUE/FALSE), however, further investigation of the selection of labeled instances is planned as further work, since there is a signal that random selection provides more gains. The findings are consistent across algorithm families (DE and PSO), underscoring that SSL's advantages are portfolio-agnostic. From a practical standpoint, we provide a simple recipe: employ RF+ELA to remain dominant only in distributionally aligned regimes, while when encountering more distribution-shifted regimes employ Doe2Vec pre-trained with SCARF, and fine-tune with at least 50-75% labels. Overall, SSL reshapes the cost–quality Pareto frontier: with modest labeling (50–75%), one can recover or surpass supervised baselines while saving a large fraction of evaluations, especially under distribution shift. The strict zero-shot case (P) remains non-trivial for all methods. This points to two promising directions: (i) masked-feature reconstruction or contrastive views that simulate class-level shifts, and (ii) richer problem-space augmentation to broaden pre-training coverage. Limitations include the focus on fixed problem dimensions/budget and selected portfolios. This study focuses on 10-dimensional single-objective continuous optimization problems. Although much higher-dimensional and large-scale problems are possible, the algorithm portfolio considered here was not designed for such settings, where the performance of its constituent algorithms is known to degrade. Furthermore, TransOptAS, DoE2Vec, and DeepELA are predefined models that were originally trained on $10d$ problems. Extending them to higher dimensions would require separate pre-training for each dimensionality, which in turn necessitates generating a substantial number of new benchmark instances using the same generator. We therefore acknowledge the restriction to $10d$ problems as a limitation of this work. Future work will explore higher-dimensional problems and additional large-scale optimizers. We do not design or evaluate domain-specific SSL pretext tasks in this work, as our aim is to assess whether existing SSL approaches can reduce labeling requirements for optimizer performance prediction in a manner that is comparable across landscape representations. Introducing custom, landscape-specific pretext tasks would reduce experimental control, make cross-representation comparisons less meaningful, and shift the focus away from the central question of label efficiency. This choice also ensures that any gains we observe arise from the SSL methods themselves rather than from domain-specific tailoring, thereby making our conclusions more general. We consider landscape-aware SSL tasks a promising direction for future work, with the potential to further improve representation quality and predictive performance. In addition, future work could explicitly quantify the wall-clock time and energy savings achieved through reduced labeling requirements and fewer function evaluations, providing a more comprehensive assessment of the practical efficiency benefits of SSL in this setting.

**LLM usage:** LLMs were used for grammar editing and minor coding support (e.g., debugging, snippets). All scientific content, design, and final code were produced and validated by the authors, who take full responsibility. **Reproducibility statement** Complete code available at https://anonymous.4open.science/r/optissl-iclr-F776. Implementation details presented in Appendix B and sensitivity study in Appendix C.

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

# A   MAE, MSE, AND RMSE AVERAGED OVER ALL FIVE DE CONFIGURATIONS

Figure 3 reports the mean absolute error (MAE) averaged over five DE configurations, shown separately for the four feature representations across evaluation splits (I, R, PC, P) under stratified label selection (RLS = TRUE). Results for mean square error (MSE) and root mean square error (RMSE) are following similar trends, and are presented in Figure 4 and Figure 5, respectively.

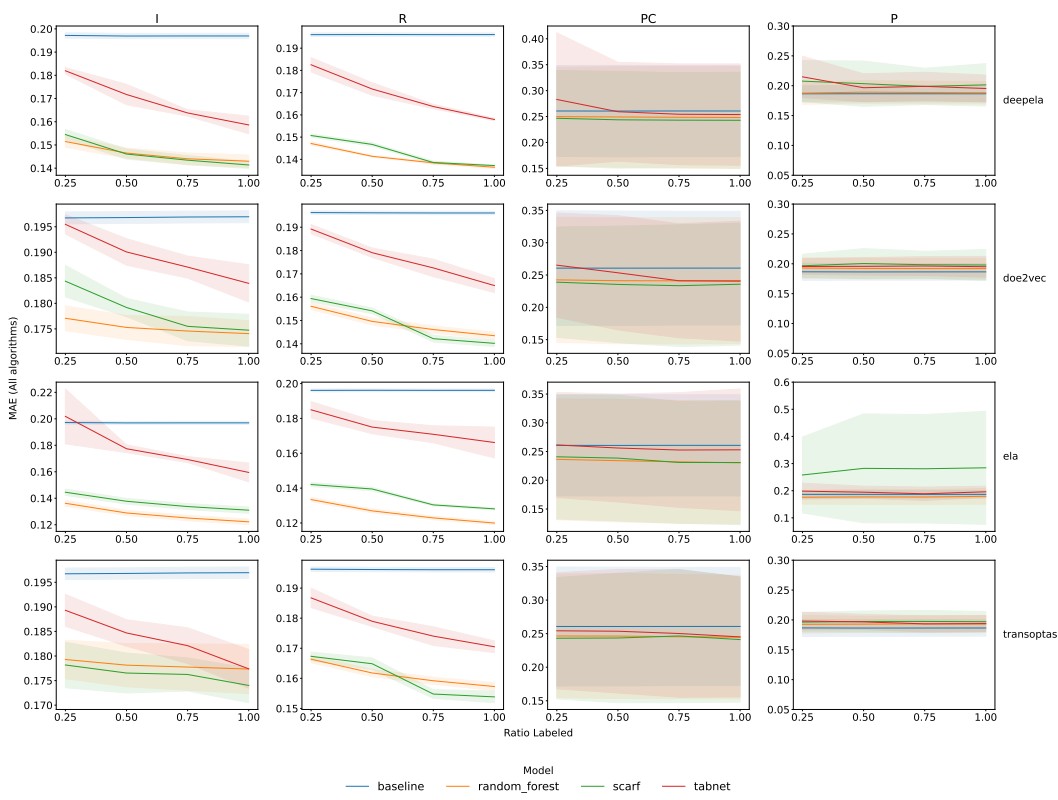

Figure 3: Mean Absolute Error (MAE) averaged over all five DE configurations, presented separately for each of the four feature representations, across the data–evaluation splits (I, R, PC, and P) using the stratified selection of labeled instances with respect to the data–evaluation split, RLS = TRUE.

# B   IMPLEMENTATION DETAILS

In terms of the training protocol, we used and adapted the implementations available on GitHub for TabNet[1] and SCARF[2], and the scikit-learn available implementation for RandomForest[3]. For pre-training TabNet, we use the official *TabNetPretrainer* on the union of labeled and unlabeled data. We then fine-tune a *TabNetRegressor* on the available labeled data, which is further randomly split into 80/20 train/validation sets, where the validation set is used for early stopping based on MSE. For pre-training SCARF, we follow the reference implementation from the GitHub repository, training the encoder for a maximum of 100 epochs. For fine-tuning, we attach a regression head on top of the pre-trained encoder, which is a simple MLP consisting of 2 hidden layers (as instructed in the respective paper) with early-stopping implemented. The fine-tuning uses a labeled 80/20

---

[1] https://github.com/dreamquark-ai/tabnet
[2] https://github.com/clabrugere/pytorch-scarf
[3] https://scikit-learn.org/1.6/modules/generated/sklearn.ensemble.RandomForestRegressor.html#sklearn.ensemble.RandomForestRegressor

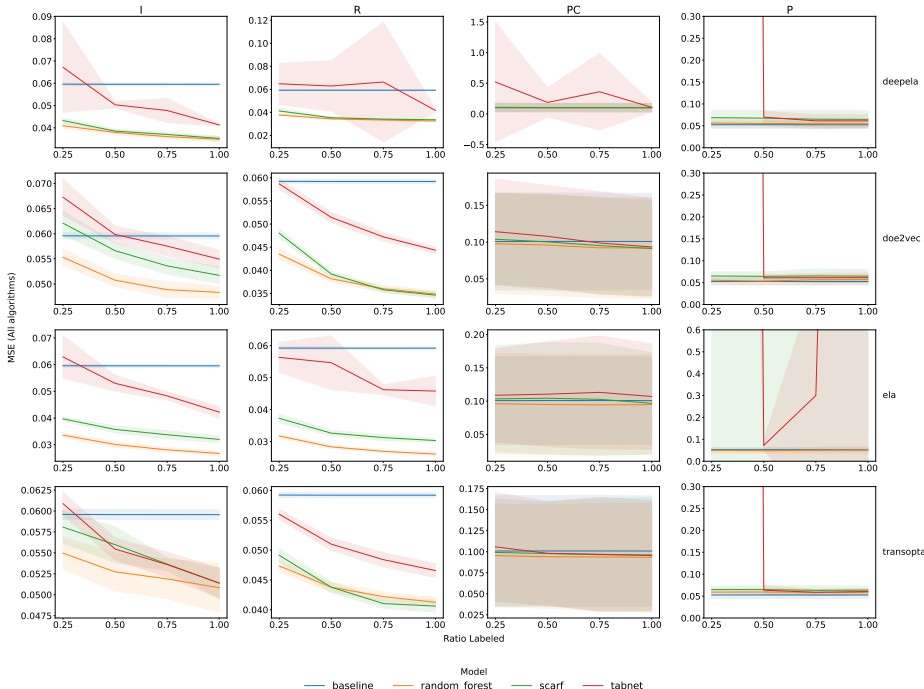

Figure 4: MSE averaged over all five DE configurations, presented separately for each of the four feature representations, across the data–evaluation splits (I, R, PC, and P) using the stratified selection of labeled instances with respect to the data–evaluation split, RLS = FALSE.

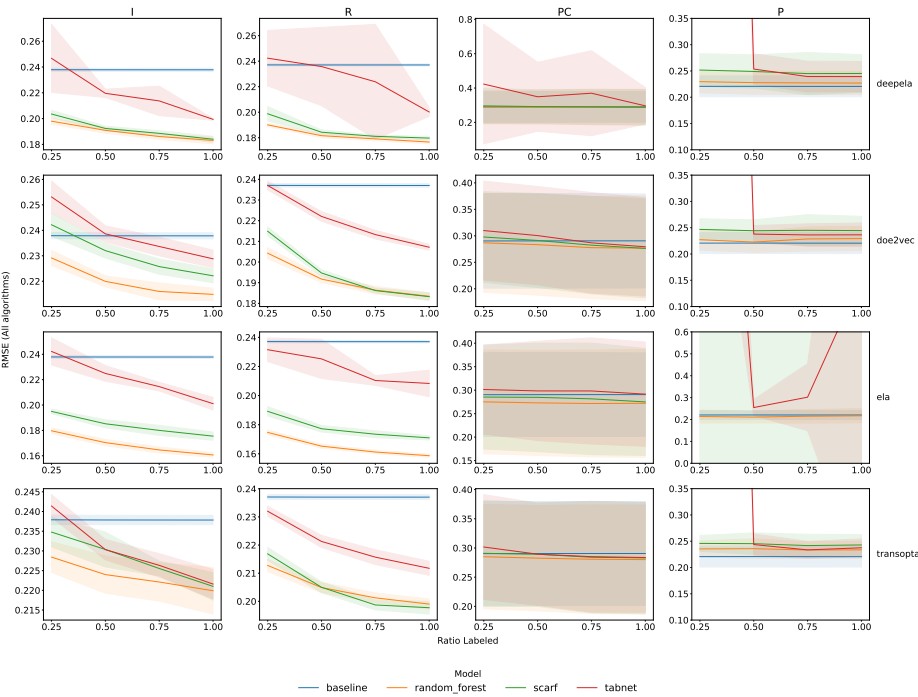

Figure 5: RMSE averaged over all five DE configurations, presented separately for each of the four feature representations, across the data–evaluation splits (I, R, PC, and P) using the stratified selection of labeled instances with respect to the data–evaluation split, RLS = FALSE.

random train/validation split, batch size of 256 and up to 50 epochs with early-stopping (patience is 10 epochs) on validation MSE. Unless stated otherwise, for both SCARF and TabNet we use the default hyperparameters, as both respective papers claim that their models are not sensitive to hyperparameter tuning (sensitivity analyses can be found in Appendix C). To maintain parity, we also use the default hyperparameters for RF. For VIME [4] and SAINT[5] we adapted the official GitHub implementations. In VIME's case, we followed the original self-supervised procedure and reimplemented it in PyTorch to integrate it into our experimental pipeline. We adhered closely to the official codebase for SAINT. We used the standard implementations of XGBoost (training as multi-output model) and LightGBM (wrapped in the `MultiOutputRegressor` [6]). All experiments are conducted on an HPC cluster. Specifically, we use a system equipped with two NVIDIA V100S GPUs, (32 GB of VRAM each), a 12-core AMD EPYC 7272 processor (24 threads) running at 2.9 GHz, 128 GB of RAM.

## C  SENSITIVITY ANALYSES OF TABNET AND SCARF

We assess the sensitivity of SCARF and TabNet to their main hyperparameters through ablation studies. The hyperparameters and their tested values are selected based on the respective original papers. Experiments are performed on the DE portfolio, using two representative feature sets: statistical ELA and learned TransOptAs features, under the R (low shift) and PC (moderate shift) splits, with stratified selection of labeled instances RLS = FALSE. We average MAE over all folds and random seeds (as outlined in the main text), and report outcomes separately for each labeled ratio.

### C.1  SCARF ABLATIONS

For SCARF, we vary three hyperparameters: corruption rate $\in \{0.2, 0.4, 0.6, 0.8\}$, batch size $\in \{64, 128, 256, 512\}$, and temperature $\in \{0.1, 1, 10\}$. Across all settings, SCARF remains stable, with minimal MAE fluctuation. The default hyperparameter configuration (corruption=0.6, batch size=256, and temperature=1) lies within this stable region.

**Corruption Rate Ablation.**  We observe that SCARF is stable across different values for the corruption rate (Figure 6). All tested values produce almost identical MAE curves, with only a marginal advantage for lower corruption rates in the PC+TransOptAs scenario.

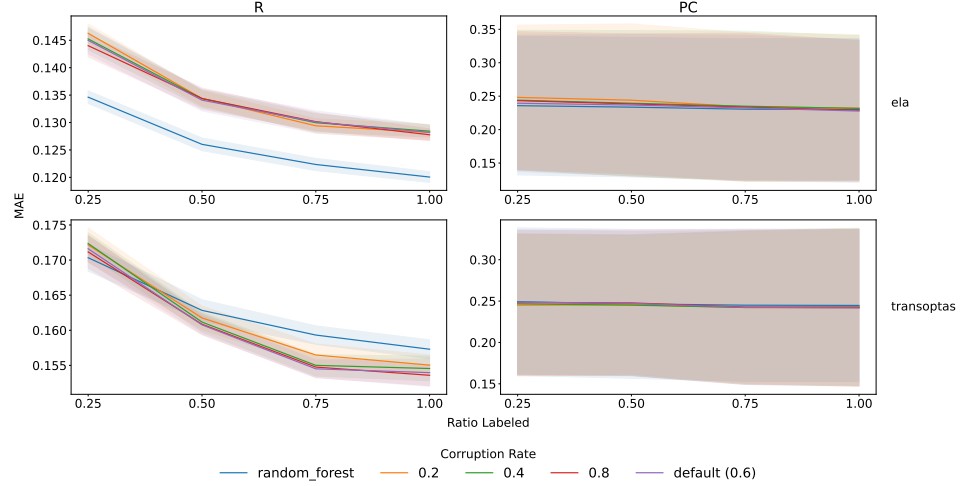

Figure 6: SCARF corruption rate ablation. MAE averaged across all five DE configurations, with the RF baseline as reference, shows the model's robustness to this hyperparameter.

---

[4]`https://github.com/jsyoon0823/VIME`
[5]`https://github.com/somepago/saint`
[6]`https://scikit-learn.org/1.6/modules/generated/sklearn.multioutput.MultiOutputRegressor.html`

**Temperature Ablation.** Figure 7 shows that smaller values of SCARF's temperature parameter can be beneficial for the ELA feature set, although not enough to change the position of SCARF relative to the RF baseline. In the TransOptAs landscape, we observe very little sensitivity to the temperature, with the default value performing as well as or better than the alternatives.

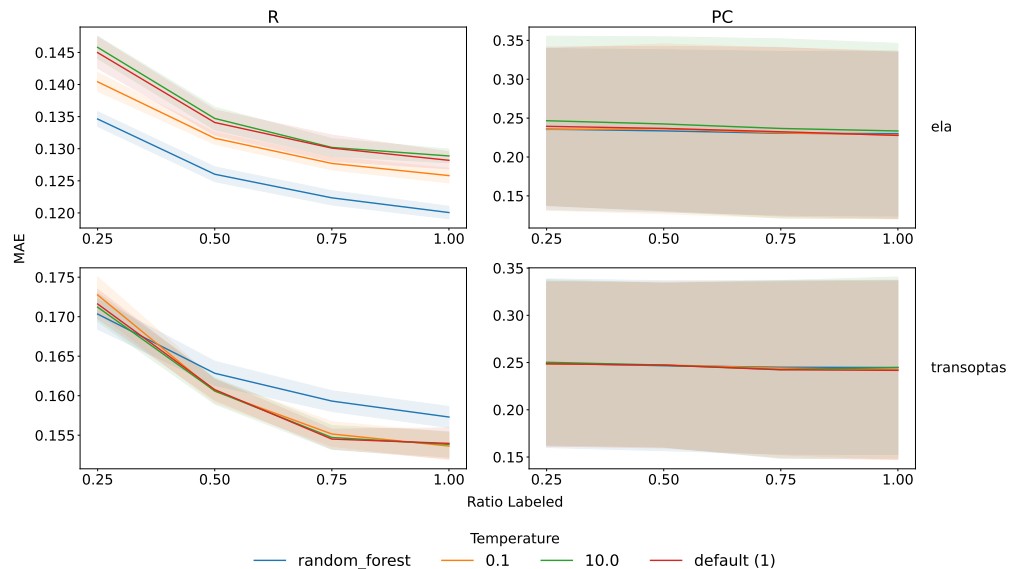

Figure 7: SCARF temperature ablation. MAE averaged across all five DE configurations, with the RF baseline included for reference. Slight variations occur, but do not change the model's position relative to RF.

**Batch Size Ablation.** Regarding the sensitivity to the batch size (Figure 8), we observe that the default choice of 256 performs competitively in all cases. In the R split, smaller batches yield slightly better performance, while in the PC+TransOptAs setting, we can get improved performance with larger batch sizes.

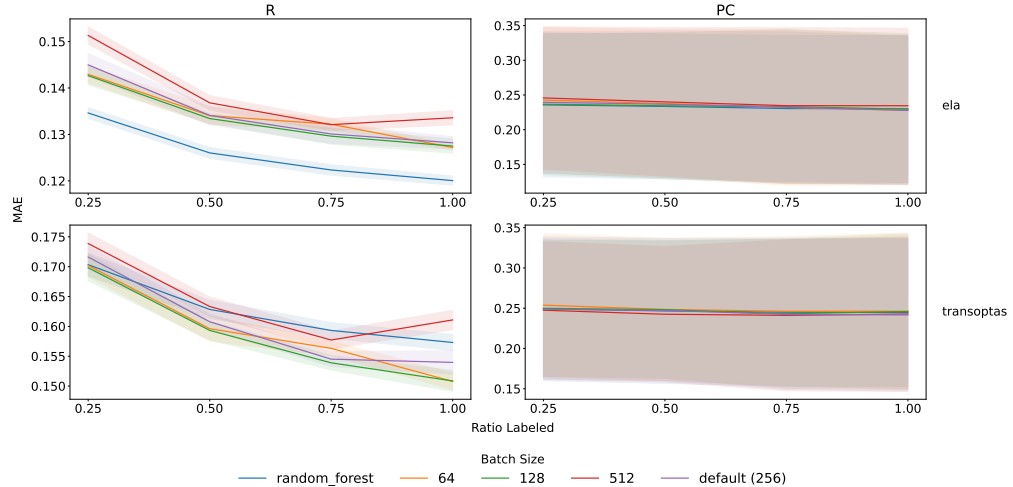

Figure 8: SCARF batch-size ablation. MAE across all five DE configurations with RF included as a reference, showcasing the model's robustness to different batch sizes.

## C.2 TABNET ABLATIONS

We performed an analogous sensitivity analysis for TabNet, varying the number of decision steps ($n_{steps} \in \{3, 5, 7, 9\}$, default=3), the prediction and attention dimensions ($n_d = n_a \in \{8, 32, 128\}$, default=8), the $\gamma$ feature reusage coefficient ($\gamma \in \{1.0, 1.3, 1.5, 2\}$, default=1.3), the sparsity loss coefficient $\lambda$ ($\lambda \in \{0.0, \text{1e-6}, \text{1e-3}, 0.1\}$, default=1e-3), and batch size $\in \{256, 512, 1024\}$ (default=1024). For TabNet we additionally include the P split to assess whether the hyperparameters affect its instability in the 25% label regime. TabNet's performance is more variable than SCARF, but these ablations do not alter the main findings of the paper.

**Number of Decision Steps Ablation.** The number of decision steps has a noticeable impact on TabNet's performance (Figure 9). Across most settings, higher step counts often leads to larger errors. The default choice of 3 steps is consistently among the best-performing configurations, especially evident at lower label ratios.

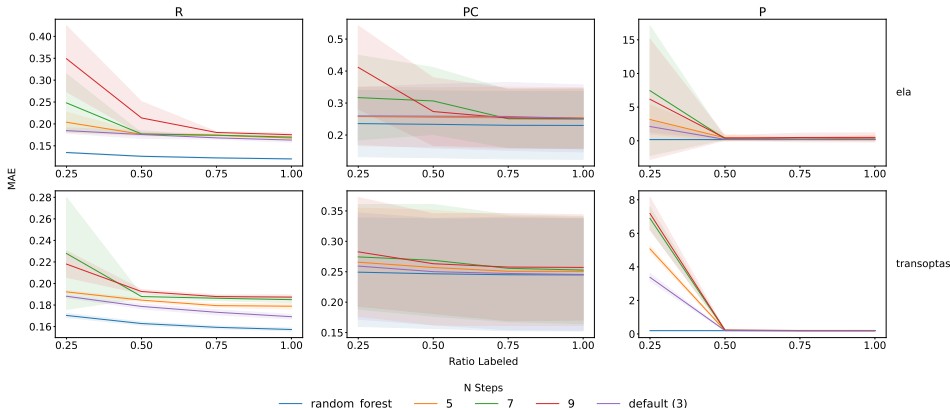

Figure 9: TabNet number-of-decision-steps ablation. MAE averaged over all five DE configurations, showing the impact of varying the number of sequential decision steps, with RF as baseline.

**Prediction and Attention Dimension Ablation.** The dimensions of the decision and attention layers control the capacity of the encoder, and are usually set to the same value. As shown in Figure 10, varying these dimensions has a modest effect on the performance of TabNet. The default value of 8 remains competitive in nearly all scenarios, and these variations do not change the conclusion of the main study.

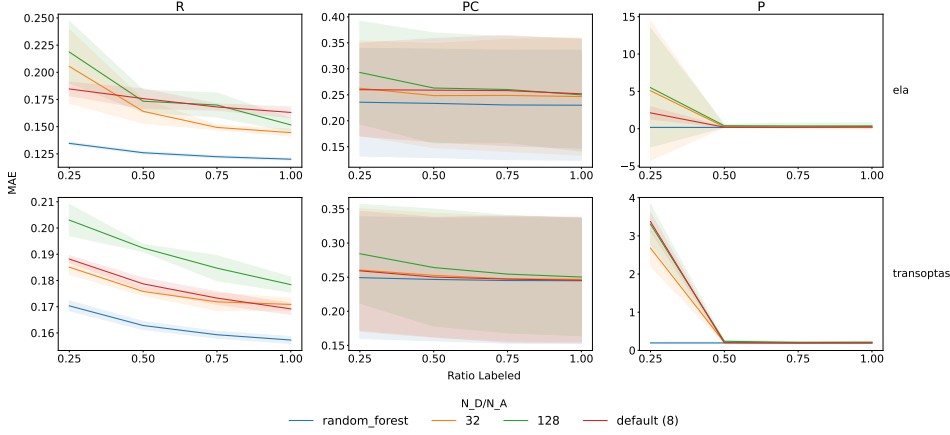

Figure 10: TabNet encoder-width ablation (number of decision/attention layers). MAE averaged over all five DE configurations, with the RF as baseline.

**Feature Reusage Coefficient ($\gamma$) Ablation.** The $\gamma$ coefficient has only a small effect on TabNet's performance (Figure 11). Across all tested scenarios, the MAE curves for different $\gamma$ show very minimal variation. These results indicate that TabNet's feature selection dynamics have little influence on the performance in this setup.

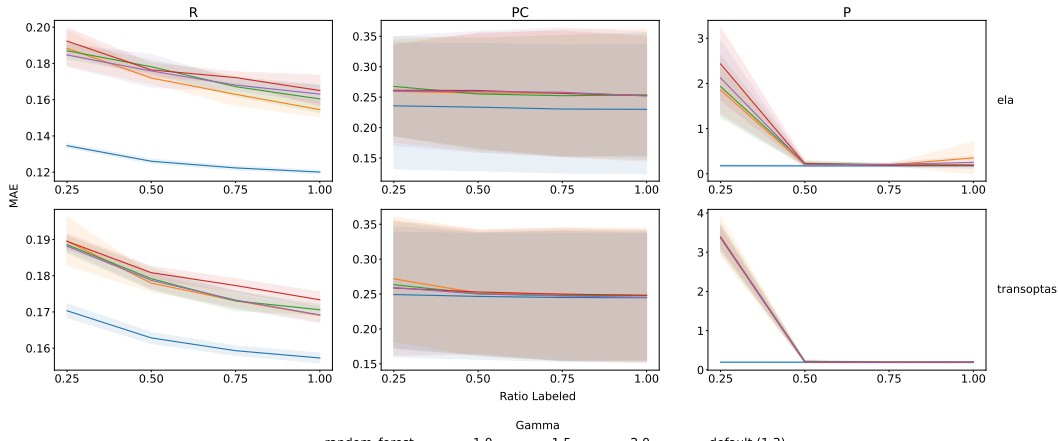

Figure 11: TabNet feature reusage coefficient $\gamma$. MAE averaged over all five DE configurations, with RF as baseline, showing little performance variation for different $\gamma$ values.

**Sparsity Coefficient ($\lambda$) Ablation.** Figure 12 shows that higher values of the sparsity regularization coefficient $\lambda$ (used only in the fine-tuning phase) slightly degrade the performance in several settings, suggesting that aggressive sparsity penalization does not help in this setting. Overall, the default value (0.001) performs competitively across scenarios, and varying $\lambda$ does not change the conclusions of the main study.

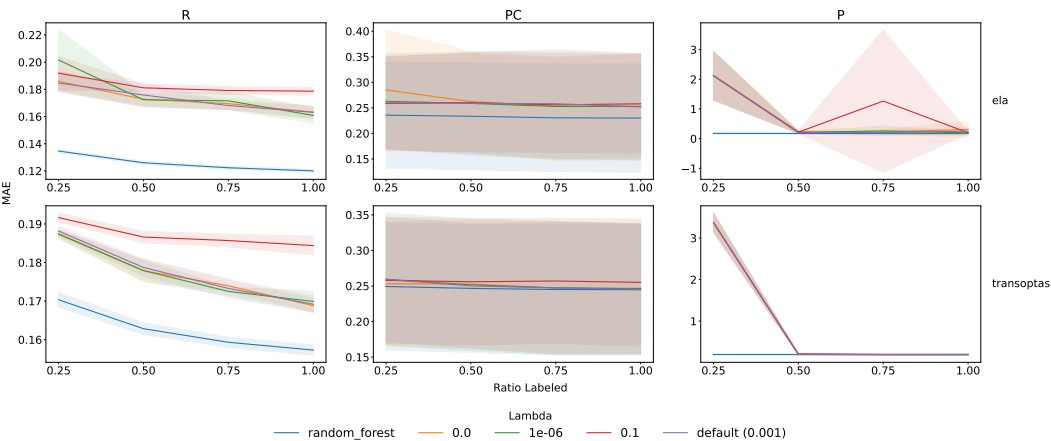

Figure 12: TabNet sparsity regularization coefficient $\lambda$. MAE averaged over all five DE configurations, compared to the RF baseline, indicating sparseness has little effect on performance.

**Batch Size Ablation.** Across most settings, TabNet's performance is weakly sensitive to the batch size (Figure 13a). The one exception is the 25% labeled setting in the P split, where smaller batch sizes allow the fine-tuning model to learn from the labeled data, instead of relying only on the pretrained encoder for the prediction. Notably, even with smaller batch sizes, TabNet still does not outperform the supervised baseline (Figure 13b), so the conclusions of the main text remain the same.

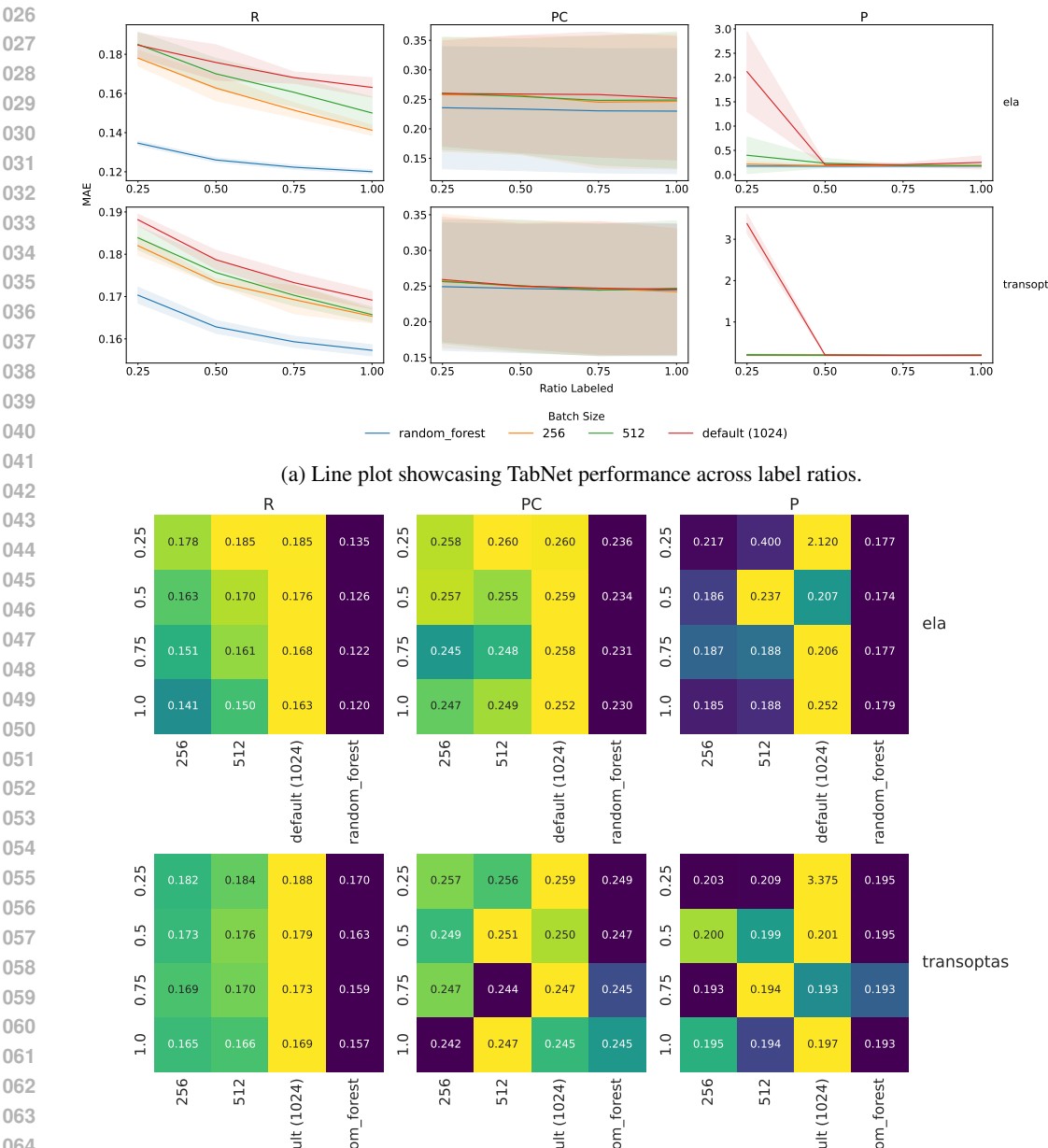

(a) Line plot showcasing TabNet performance across label ratios.

(b) Heatmap showing MAE across different batch sizes for TabNet (and the RF baseline); The ratio of labeled examples is on the $y$ axis. Darker values for each row indicate lower values for MAE.

Figure 13: TabNet batch-size ablation. MAE averaged over all five DE configurations, with RF as baseline.

# D EXPERIMENTS WITH ADDITIONAL SUPERVISED AND SELF-SUPERVISED MODELS

To further contextualize the results in the main text, we evaluate additional supervised and self-supervised models under the same experimental protocol, focusing on the DE portfolio with RLS = False.

### D.1 ADDITIONAL SUPERVISED MODELS

To strengthen our claims regarding the supervised baselines, we additionally evaluated two widely-used gradient boosted tree models: XGBoost and LightGBM. The results are presented in Figure 14 (with SCARF added for easier comparison). Overall, the boosted trees do not provide meaningful improvements over RF, or substantial changes in the conclusions in the main paper. LightGBM closely trails RF, while XGBoost performs worse than RF in most settings.

In the low-shift splits (I and R), LightGBM shows very similar performance to RF, with slightly better performance in the R+DoE2Vec combination; XGBoost is the weakest performer, particularly when using learned representations (TransOptAs, Doe2Vec, DeepELA). In the PC and P splits, the performance gap between the tree-based baselines narrows, rarely improving over RF.

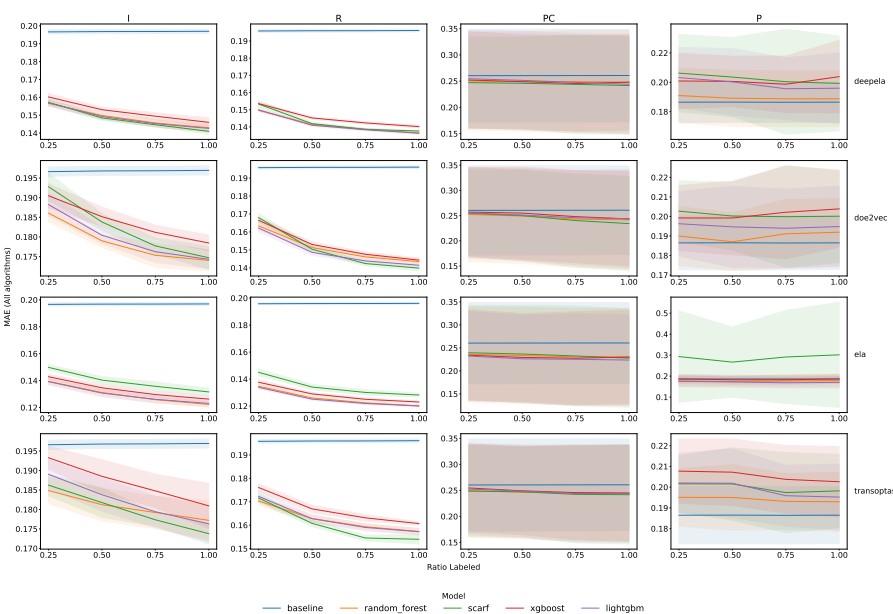

Figure 14: Supervised model comparison (DE, RLS = FALSE). MAE averaged over all DE configurations for RF, XGBoost, LightGBM, and SCARF (included for reference).

### D.2 ADDITIONAL SELF-SUPERVISED MODELS

We evaluated two additional SSL methods, VIME and SAINT, to complement SCARF and TabNet and provide a broader perspective on self-supervised learning in this scenario. SAINT is a representative of the third SSL paradigm, hybrid SSL (a combination of predictive and contrastive pretext tasks), and VIME as a second representative of the predictive SSL paradigm, allowing us to assess whether the weak performance of TabNet is model-specific or reflects broader limitations of predictive SSL in this domain. The results are presented in Figure 15. Across all settings, VIME and SAINT exhibit performance that places them between SCARF and TabNet, with the exception of P+ELA scenario, where SAINT is the best SSL performer, but still does not outperform the supervised baselines. Incorporating these additional SSL models reinforces the conclusions in the main text.

## E   SSL VS. ORIGINAL EMBEDDING SIMILARITY

To examine what SSL embeddings capture beyond the original representations, we identified for each test instance its most similar labeled training instance and computed their cosine similarity using both the original features (ELA, DeepELA, and DoE2Vec) and the SSL-derived embeddings, along with their absolute performance difference. This in-distribution (I) analysis, run across all label ratios on a single illustrative fold with RLS = False, directly tests whether SSL makes similarity more reflective of true performance behavior.

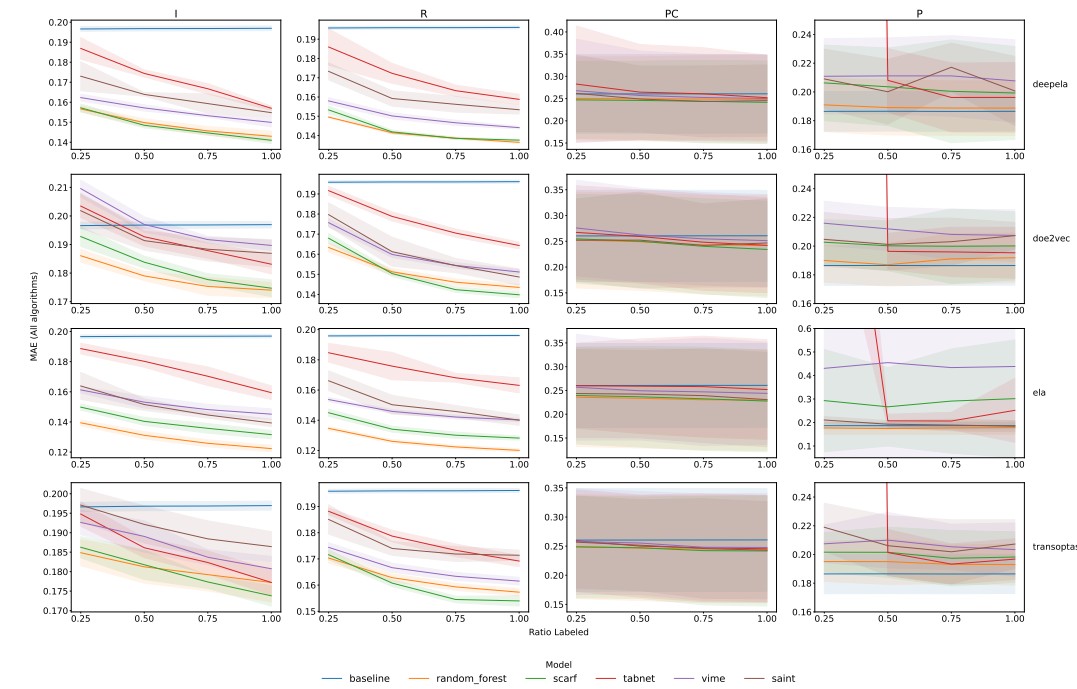

Figure 15: Performance of additional SSL methods. MAE across all five DE configurations for SCARF, VIME, SAINT, and TabNet. VIME and SAINT perform between SCARF and TabNet, without surpassing supervised baselines.

At a 75% training ratio, the DeepELA plots show how similarity to the closest training instance relates to absolute performance difference (Figure 16). The x-axis shows the similarity between each test instance and its closest training instance, while the y-axis shows how much their true performance differs. A good representation places performance-similar problems close together, resulting in many points with high similarity and low performance difference clustered in the bottom-right corner. In the baseline DeepELA space, similarities range widely (0.65–1.00), and even highly similar instances (>0.9) often show large performance gaps (>0.60), indicating a noisy and weak alignment between similarity and true performance. DeepELA + TabNet collapses similarity into a very narrow band near 1.0 (0.99–1.00), eliminating meaningful resolution. Large performance differences still appear within this saturated region, making similarity uninformative—mirroring the behavior seen in TransOptAS + TabNet. DeepELA + SCARF shows clear improvement: similarities shift upward (0.88–1.00), forming a dense cluster at very high similarity (0.98–1.00) with low performance differences (0.00–0.20) and far fewer mismatches. SCARF reshapes the latent space so similarity reliably reflects performance, suppressing noise and sharpening structure. Overall, baseline DeepELA is weakly aligned, TabNet destroys discriminative similarity, and SCARF is the only method that makes similarity meaningfully correspond to performance differences.

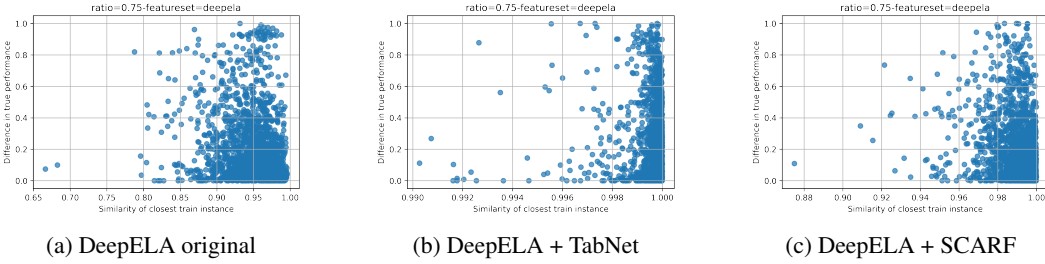

(a) DeepELA original    (b) DeepELA + TabNet    (c) DeepELA + SCARF

Figure 16: The post-hoc analysis of whether SSL reshapes the DeepELA feature space so that similarity more accurately corresponds to true performance similarity.

For ELA at a 75% training ratio, the original ELA representation yields the best behavior: although noisy, it maintains a reasonable spread of similarity values (0.60–1.00) and forms a visible cluster of high-similarity, low-difference pairs, meaning that similarity still carries some performance-relevant signal (Figure 17). In contrast, SCARF and TabNet do not improve ELA. SCARF shifts similarities upward but does not significantly reduce the number of large performance gaps among highly similar instances. TabNet collapses similarity almost entirely into a narrow band near 1.00, removing meaningful resolution while still producing large performance differences. Thus, for ELA, the original representation provides the strongest link between similarity and true performance, and SSL methods neither enhance nor preserve this structure.

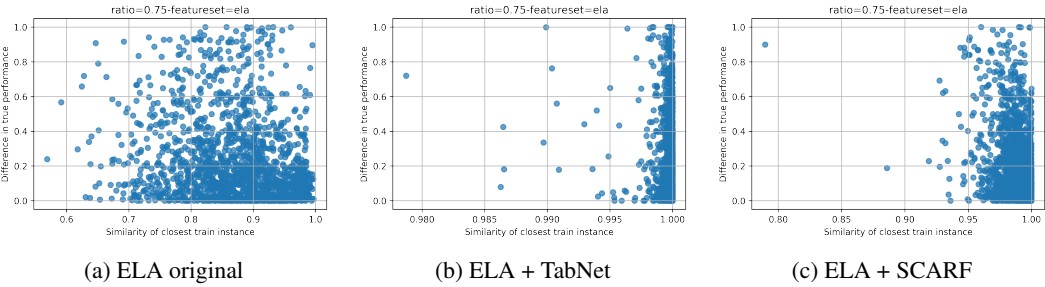

(a) ELA original        (b) ELA + TabNet        (c) ELA + SCARF

Figure 17: The post-hoc analysis of whether SSL reshapes the ELA feature space so that similarity more accurately corresponds to true performance similarity.

For DoE2Vec at a 75% training ratio (Figure 18), the original representation and SCARF produce very similar outcomes. Both maintain a broad spread of similarity values and show a large cluster of high-similarity, low-difference pairs, while still exhibiting some noisy high-difference points. This indicates that DoE2Vec already captures performance-relevant structure reasonably well, and SCARF does not drastically alter or improve this alignment, but importantly, it does not degrade it either. In contrast, TabNet collapses similarity into a narrow region near 1.00, removing meaningful resolution while still producing large performance gaps, making similarity uninformative. Thus, for DoE2Vec, the original and SCARF-enhanced spaces behave similarly, whereas TabNet distorts the structure.

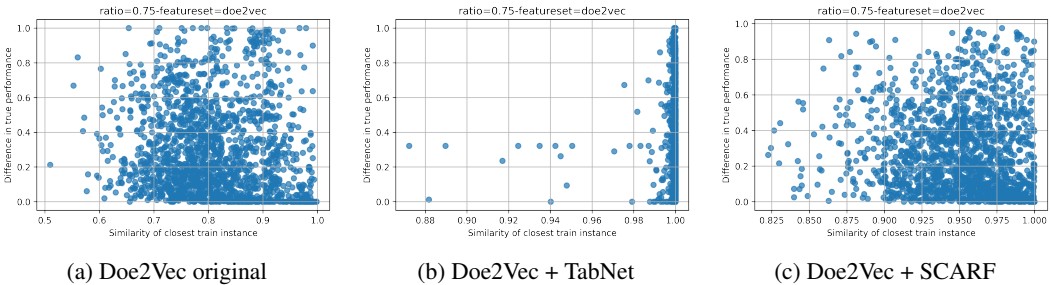

(a) Doe2Vec original        (b) Doe2Vec + TabNet        (c) Doe2Vec + SCARF

Figure 18: The post-hoc analysis of whether SSL reshapes the Doe2Vec feature space so that similarity more accurately corresponds to true performance similarity.

## F  MAEs for PSO and tables for omitted function evaluations

Table 3 presents the MAE (mean ± sd) aggregated across all PSO configurations for each feature representation and data–evaluation split, with/without stratified labeled instance selection, and for different ratios of labeled instances.

Table 4 reports the percentage of function evaluations that can be omitted when self-supervised learning (SCARF, TabNet) outperforms the naive baseline calculated with 100% labeled data in the case of the DE portfolio. Here we present the analysis per algorithm configuration, offering more fine-grained insights.

Table 3: Mean Absolute Error (MAE; mean ± sd) across feature representations and data–evaluation splits with/without stratified labeled instance selection (RLS) - PSO.

| | | RLS | False | | | | True | | | |
|---|---|---|---|---|---|---|---|---|---|---|
| | | Model | Baseline | RF | SCARF | TabNet | Baseline | RF | SCARF | TabNet |
| Evaluation split | Feature representation | Ratio Labeled | | | | | | | | |
| I | deepela | 0.25 | 0.134 ± 0.002 | 0.109 ± 0.003 | 0.116 ± 0.003 | 0.130 ± 0.011 | 0.136 ± 0.002 | 0.107 ± 0.003 | 0.114 ± 0.004 | 0.134 ± 0.011 |
| | | 0.50 | 0.135 ± 0.002 | 0.106 ± 0.003 | 0.107 ± 0.004 | 0.122 ± 0.003 | 0.135 ± 0.002 | **0.105 ± 0.003** | **0.105 ± 0.003** | 0.120 ± 0.004 |
| | | 0.75 | 0.135 ± 0.002 | 0.104 ± 0.003 | **0.102 ± 0.003** | 0.116 ± 0.004 | 0.135 ± 0.002 | 0.104 ± 0.003 | **0.102 ± 0.003** | 0.115 ± 0.002 |
| | | 1.00 | 0.135 ± 0.002 | 0.103 ± 0.003 | **0.101 ± 0.003** | 0.111 ± 0.003 | 0.135 ± 0.002 | 0.103 ± 0.003 | **0.101 ± 0.003** | 0.112 ± 0.003 |
| | doe2vec | 0.25 | 0.134 ± 0.002 | 0.132 ± 0.003 | 0.141 ± 0.003 | 0.145 ± 0.002 | 0.135 ± 0.002 | 0.126 ± 0.004 | 0.132 ± 0.004 | 0.138 ± 0.001 |
| | | 0.50 | 0.135 ± 0.002 | 0.127 ± 0.002 | 0.135 ± 0.004 | 0.137 ± 0.002 | 0.135 ± 0.002 | 0.125 ± 0.004 | 0.129 ± 0.005 | 0.134 ± 0.002 |
| | | 0.75 | 0.135 ± 0.002 | 0.125 ± 0.003 | 0.130 ± 0.002 | 0.132 ± 0.002 | 0.135 ± 0.002 | 0.124 ± 0.004 | 0.126 ± 0.005 | 0.132 ± 0.004 |
| | | 1.00 | 0.135 ± 0.002 | 0.124 ± 0.005 | 0.126 ± 0.004 | 0.129 ± 0.002 | 0.135 ± 0.002 | 0.124 ± 0.005 | 0.126 ± 0.005 | 0.129 ± 0.002 |
| | ela | 0.25 | 0.134 ± 0.002 | 0.099 ± 0.002 | 0.112 ± 0.002 | 0.151 ± 0.023 | 0.136 ± 0.002 | 0.097 ± 0.002 | 0.111 ± 0.002 | 0.131 ± 0.004 |
| | | 0.50 | 0.135 ± 0.002 | 0.094 ± 0.002 | 0.103 ± 0.002 | 0.125 ± 0.001 | 0.135 ± 0.002 | 0.093 ± 0.002 | 0.102 ± 0.002 | 0.120 ± 0.002 |
| | | 0.75 | 0.135 ± 0.002 | 0.092 ± 0.002 | 0.099 ± 0.002 | 0.12 ± 0.002 | 0.135 ± 0.002 | 0.091 ± 0.002 | 0.099 ± 0.002 | 0.119 ± 0.002 |
| | | 1.00 | 0.135 ± 0.002 | 0.090 ± 0.002 | 0.097 ± 0.001 | 0.114 ± 0.003 | 0.135 ± 0.002 | 0.09 ± 0.002 | 0.097 ± 0.001 | 0.114 ± 0.003 |
| | transoptas | 0.25 | 0.134 ± 0.002 | 0.134 ± 0.004 | 0.135 ± 0.003 | 0.138 ± 0.003 | 0.135 ± 0.002 | 0.128 ± 0.006 | 0.129 ± 0.006 | 0.146 ± 0.026 |
| | | 0.50 | 0.135 ± 0.002 | 0.130 ± 0.005 | **0.128 ± 0.005** | 0.132 ± 0.003 | 0.135 ± 0.002 | 0.127 ± 0.006 | **0.124 ± 0.004** | 0.130 ± 0.003 |
| | | 0.75 | 0.135 ± 0.002 | 0.128 ± 0.006 | **0.126 ± 0.003** | 0.130 ± 0.004 | 0.135 ± 0.002 | 0.127 ± 0.006 | **0.123 ± 0.005** | 0.128 ± 0.004 |
| | | 1.00 | 0.135 ± 0.002 | 0.127 ± 0.006 | **0.123 ± 0.004** | **0.125 ± 0.004** | 0.135 ± 0.002 | 0.127 ± 0.006 | **0.123 ± 0.004** | **0.125 ± 0.005** |
| R | deepela | 0.25 | 0.135 ± 0.000 | 0.106 ± 0.000 | 0.112 ± 0.0 | 0.125 ± 0.001 | / | / | / | / |
| | | 0.50 | 0.135 ± 0.000 | 0.102 ± 0.000 | 0.104 ± 0.000 | 0.118 ± 0.002 | / | / | / | / |
| | | 0.75 | 0.135 ± 0.000 | **0.100 ± 0.000** | **0.100 ± 0.000** | 0.115 ± 0.001 | / | / | / | / |
| | | 1.00 | 0.135 ± 0.000 | **0.099 ± 0.000** | **0.099 ± 0.000** | 0.112 ± 0.002 | / | / | / | / |
| | doe2vec | 0.25 | 0.135 ± 0.000 | 0.116 ± 0.000 | 0.124 ± 0.001 | 0.135 ± 0.001 | / | / | / | / |
| | | 0.50 | 0.135 ± 0.000 | 0.108 ± 0.001 | 0.11 ± 0.001 | 0.128 ± 0.001 | / | / | / | / |
| | | 0.75 | 0.135 ± 0.000 | 0.104 ± 0.001 | **0.103 ± 0.001** | 0.122 ± 0.001 | / | / | / | / |
| | | 1.00 | 0.135 ± 0.000 | 0.103 ± 0.000 | **0.100 ± 0.001** | 0.117 ± 0.002 | / | / | / | / |
| | ela | 0.25 | 0.135 ± 0.000 | 0.096 ± 0.001 | 0.108 ± 0.001 | 0.127 ± 0.003 | / | / | / | / |
| | | 0.50 | 0.135 ± 0.000 | 0.092 ± 0.000 | 0.1 ± 0.001 | 0.122 ± 0.005 | / | / | / | / |
| | | 0.75 | 0.135 ± 0.000 | 0.089 ± 0.000 | 0.096 ± 0.001 | 0.117 ± 0.003 | / | / | / | / |
| | | 1.00 | 0.135 ± 0.000 | 0.088 ± 0.000 | 0.094 ± 0.001 | 0.116 ± 0.005 | / | / | / | / |
| | transoptas | 0.25 | 0.135 ± 0.000 | 0.122 ± 0.001 | 0.124 ± 0.001 | 0.134 ± 0.002 | / | / | / | / |
| | | 0.50 | 0.135 ± 0.000 | **0.116 ± 0.001** | **0.116 ± 0.001** | 0.128 ± 0.001 | / | / | / | / |
| | | 0.75 | 0.135 ± 0.000 | 0.113 ± 0.001 | **0.111 ± 0.001** | 0.124 ± 0.001 | / | / | / | / |
| | | 1.00 | 0.135 ± 0.000 | 0.112 ± 0.001 | **0.109 ± 0.001** | 0.121 ± 0.001 | / | | | |
| PC | deepela | 0.25 | 0.201 ± 0.160 | 0.196 ± 0.146 | **0.192 ± 0.148** | 0.211 ± 0.183 | 0.200 ± 0.159 | 0.196 ± 0.144 | **0.194 ± 0.148** | 0.221 ± 0.203 |
| | | 0.50 | 0.201 ± 0.160 | 0.191 ± 0.146 | **0.187 ± 0.141** | 0.197 ± 0.161 | 0.201 ± 0.160 | 0.192 ± 0.146 | **0.189 ± 0.148** | 0.198 ± 0.159 |
| | | 0.75 | 0.201 ± 0.160 | 0.189 ± 0.146 | **0.188 ± 0.149** | 0.196 ± 0.162 | 0.201 ± 0.159 | 0.191 ± 0.144 | **0.190 ± 0.150** | 0.193 ± 0.154 |
| | | 1.00 | 0.201 ± 0.159 | 0.189 ± 0.146 | **0.186 ± 0.144** | 0.195 ± 0.156 | 0.201 ± 0.159 | 0.189 ± 0.146 | 0.192 ± 0.148 | 0.195 ± 0.159 |
| | doe2vec | 0.25 | 0.201 ± 0.160 | 0.210 ± 0.146 | 0.205 ± 0.148 | 0.214 ± 0.154 | 0.200 ± 0.161 | 0.190 ± 0.145 | **0.189 ± 0.145** | 0.205 ± 0.160 |
| | | 0.50 | 0.201 ± 0.160 | 0.202 ± 0.150 | **0.200 ± 0.149** | 0.204 ± 0.159 | 0.201 ± 0.160 | 0.189 ± 0.145 | **0.185 ± 0.147** | 0.197 ± 0.157 |
| | | 0.75 | 0.201 ± 0.160 | 0.192 ± 0.146 | **0.191 ± 0.150** | **0.191 ± 0.158** | 0.201 ± 0.159 | 0.188 ± 0.145 | **0.183 ± 0.146** | 0.193 ± 0.159 |
| | | 1.00 | 0.201 ± 0.159 | 0.188 ± 0.145 | **0.184 ± 0.147** | 0.190 ± 0.153 | 0.201 ± 0.159 | 0.188 ± 0.144 | **0.181 ± 0.145** | 0.188 ± 0.152 |
| | ela | 0.25 | 0.201 ± 0.160 | 0.187 ± 0.154 | **0.186 ± 0.159** | 0.204 ± 0.156 | 0.200 ± 0.159 | **0.187 ± 0.150** | **0.187 ± 0.154** | 0.201 ± 0.155 |
| | | 0.50 | 0.201 ± 0.160 | 0.185 ± 0.152 | 0.186 ± 0.158 | 0.207 ± 0.156 | 0.201 ± 0.160 | 0.184 ± 0.151 | **0.182 ± 0.149** | 0.193 ± 0.162 |
| | | 0.75 | 0.201 ± 0.160 | 0.181 ± 0.151 | 0.184 ± 0.162 | 0.189 ± 0.162 | 0.201 ± 0.159 | 0.180 ± 0.150 | 0.184 ± 0.150 | 0.192 ± 0.165 |
| | | 1.00 | 0.201 ± 0.159 | 0.182 ± 0.148 | **0.179 ± 0.15** | 0.189 ± 0.159 | 0.201 ± 0.159 | 0.181 ± 0.148 | **0.179 ± 0.154** | 0.191 ± 0.165 |
| | transoptas | 0.25 | 0.201 ± 0.160 | 0.193 ± 0.138 | **0.189 ± 0.142** | 0.203 ± 0.143 | 0.200 ± 0.161 | 0.189 ± 0.146 | **0.186 ± 0.154** | 0.199 ± 0.152 |
| | | 0.50 | 0.201 ± 0.160 | 0.193 ± 0.144 | **0.189 ± 0.150** | 0.196 ± 0.152 | 0.201 ± 0.160 | 0.189 ± 0.146 | **0.186 ± 0.156** | 0.195 ± 0.147 |
| | | 0.75 | 0.201 ± 0.160 | 0.190 ± 0.147 | **0.189 ± 0.157** | 0.191 ± 0.154 | 0.201 ± 0.159 | 0.189 ± 0.146 | **0.188 ± 0.159** | 0.190 ± 0.149 |
| | | 1.00 | 0.201 ± 0.159 | 0.189 ± 0.146 | **0.186 ± 0.158** | 0.190 ± 0.152 | 0.201 ± 0.159 | 0.189 ± 0.146 | **0.183 ± 0.157** | 0.188 ± 0.152 |
| P | deepela | 0.25 | 0.162 ± 0.030 | 0.156 ± 0.029 | 0.161 ± 0.034 | 2.908 ± 0.459 | 0.162 ± 0.030 | 0.156 ± 0.028 | 0.158 ± 0.033 | 0.218 ± 0.140 |
| | | 0.50 | 0.162 ± 0.030 | 0.153 ± 0.026 | 0.157 ± 0.032 | 0.167 ± 0.034 | 0.162 ± 0.030 | 0.153 ± 0.027 | 0.158 ± 0.032 | 0.161 ± 0.0320 |
| | | 0.75 | 0.162 ± 0.030 | 0.153 ± 0.027 | 0.160 ± 0.031 | 0.159 ± 0.032 | 0.162 ± 0.030 | 0.153 ± 0.027 | 0.156 ± 0.031 | 0.159 ± 0.034 |
| | | 1.00 | 0.162 ± 0.030 | 0.153 ± 0.027 | 0.157 ± 0.031 | 0.158 ± 0.030 | 0.162 ± 0.030 | 0.153 ± 0.027 | 0.156 ± 0.030 | 0.159 ± 0.035 |
| | doe2vec | 0.25 | 0.162 ± 0.030 | 0.162 ± 0.019 | 0.172 ± 0.025 | 2.99 ± 0.279 | 0.162 ± 0.030 | 0.158 ± 0.023 | 0.165 ± 0.026 | 0.167 ± 0.028 |
| | | 0.50 | 0.162 ± 0.030 | 0.159 ± 0.022 | 0.172 ± 0.028 | 0.167 ± 0.026 | 0.162 ± 0.030 | 0.159 ± 0.0250 | 0.166 ± 0.028 | 0.164 ± 0.027 |
| | | 0.75 | 0.162 ± 0.030 | 0.158 ± 0.024 | 0.168 ± 0.028 | 0.165 ± 0.030 | 0.162 ± 0.030 | 0.158 ± 0.026 | 0.164 ± 0.028 | 0.163 ± 0.031 |
| | | 1.00 | 0.162 ± 0.030 | 0.157 ± 0.025 | 0.165 ± 0.027 | 0.163 ± 0.029 | 0.162 ± 0.030 | 0.157 ± 0.025 | 0.163 ± 0.028 | 0.161 ± 0.027 |
| | ela | 0.25 | 0.162 ± 0.030 | 0.151 ± 0.021 | 0.210 ± 0.135 | 2.594 ± 1.055 | 0.162 ± 0.030 | 0.148 ± 0.021 | 0.203 ± 0.121 | 0.243 ± 0.178 |
| | | 0.50 | 0.162 ± 0.030 | 0.146 ± 0.018 | 0.205 ± 0.138 | 0.195 ± 0.100 | 0.162 ± 0.030 | 0.147 ± 0.020 | 0.223 ± 0.171 | 0.170 ± 0.046 |
| | | 0.75 | 0.162 ± 0.030 | 0.144 ± 0.018 | 0.254 ± 0.243 | 0.162 ± 0.034 | 0.162 ± 0.030 | 0.145 ± 0.019 | 0.218 ± 0.162 | 0.162 ± 0.025 |
| | | 1.00 | 0.162 ± 0.030 | 0.144 ± 0.019 | 0.214 ± 0.166 | 0.155 ± 0.029 | 0.162 ± 0.030 | 0.145 ± 0.019 | 0.233 ± 0.202 | 0.157 ± 0.035 |
| | transoptas | 0.25 | 0.162 ± 0.030 | 0.166 ± 0.032 | 0.173 ± 0.032 | 3.571 ± 0.389 | 0.162 ± 0.030 | 0.164 ± 0.03 | 0.165 ± 0.030 | 0.168 ± 0.034 |
| | | 0.50 | 0.162 ± 0.030 | 0.165 ± 0.032 | 0.167 ± 0.034 | 0.168 ± 0.034 | 0.162 ± 0.030 | 0.165 ± 0.030 | 0.165 ± 0.032 | 0.166 ± 0.031 |
| | | 0.75 | 0.162 ± 0.030 | 0.164 ± 0.029 | 0.165 ± 0.032 | 0.165 ± 0.031 | 0.162 ± 0.030 | 0.165 ± 0.029 | 0.164 ± 0.029 | 0.164 ± 0.031 |
| | | 1.00 | 0.162 ± 0.030 | 0.165 ± 0.030 | 0.165 ± 0.033 | 0.164 ± 0.033 | 0.162 ± 0.030 | 0.164 ± 0.030 | 0.168 ± 0.034 | 0.163 ± 0.029 |

Table 4: Aggregated efficiency gains of self-supervised learning for DE, reported as the percentage of function evaluations that can be omitted ($\Delta_{f,s,\ell,r}$) when SCARF or TabNet achieves lower MAE than the naive baseline calculated with 100% labeled ground performance data. Results are averaged across five folds and five repetitions for each configuration of feature representation, evaluation split (I, R, PC, P), and stratification strategy (RLS = FALSE: split-aligned, RLS = TRUE: random). Label ratios of 25%, 50%, and 75% are considered, while the 100% case is omitted since all instances must be labeled. Higher values indicate greater savings in ground-truth evaluations.

| Evaluation split | | I | | | | | R | | | | | PC | | | | | P | | | | |
|---|---|---|---|---|---|---|---|---|---|---|---|---|---|---|---|---|---|---|---|---|---|
| | | DE1 | DE2 | DE3 | DE4 | DE5 | DE1 | DE2 | DE3 | DE4 | DE5 | DE1 | DE2 | DE3 | DE4 | DE5 | DE1 | DE2 | DE3 | DE4 | DE5 |
| RLS | % labeled | | | | | | | | | | | | | | | | | | | | |
| **DeepELA** | | | | | | | | | | | | | | | | | | | | | |
| True | 0.25 | 100 | 100 | 100 | 100 | 100 | / | / | / | / | / | 60 | 60 | 80 | 40 | 60 | 0 | 80 | 40 | 0 | 20 |
| | 0.5 | 100 | 100 | 100 | 100 | 100 | / | / | / | / | / | 80 | 60 | 80 | 60 | 60 | 20 | 80 | 80 | 20 | 20 |
| | 0.75 | 100 | 100 | 100 | 100 | 100 | / | / | / | / | / | 80 | 60 | 80 | 60 | 60 | 20 | 80 | 80 | 0 | 20 |
| False | 0.25 | 100 | 100 | 100 | 100 | 100 | 100 | 100 | 100 | 100 | 100 | 43 | 64 | 85 | 43 | 64 | 0 | 85 | 85 | 0 | 0 |
| | 0.5 | 100 | 100 | 100 | 100 | 100 | 100 | 100 | 100 | 100 | 100 | 72 | 72 | 96 | 48 | 96 | 0 | 96 | 72 | 0 | 48 |
| | 0.75 | 100 | 100 | 100 | 100 | 100 | 100 | 100 | 100 | 100 | 100 | 64 | 48 | 64 | 32 | 48 | 16 | 64 | 48 | 16 | 16 |
| **Doe2Vec** | | | | | | | | | | | | | | | | | | | | | |
| True | 0.25 | 80 | 100 | 100 | 100 | 100 | / | / | / | / | / | 40 | 100 | 100 | 60 | 40 | 20 | 60 | 40 | 20 | 0 |
| | 0.5 | 100 | 100 | 100 | 100 | 100 | / | / | / | / | / | 80 | 100 | 100 | 60 | 60 | 20 | 60 | 60 | 20 | 20 |
| | 0.75 | 100 | 100 | 100 | 100 | 100 | / | / | / | / | / | 80 | 80 | 100 | 40 | 80 | 20 | 80 | 60 | 20 | 20 |
| False | 0.25 | 40 | 100 | 80 | 80 | 40 | 100 | 100 | 100 | 100 | 100 | 21 | 43 | 100 | 64 | 0 | 0 | 43 | 43 | 0 | 0 |
| | 0.5 | 80 | 100 | 100 | 100 | 100 | 100 | 100 | 100 | 100 | 100 | 48 | 120 | 120 | 72 | 24 | 24 | 72 | 48 | 24 | 24 |
| | 0.75 | 100 | 100 | 100 | 100 | 100 | 100 | 100 | 100 | 100 | 100 | 64 | 80 | 80 | 16 | 32 | 16 | 64 | 32 | 16 | 16 |
| **ELA** | | | | | | | | | | | | | | | | | | | | | |
| True | 0.25 | 100 | 100 | 100 | 100 | 100 | / | / | / | / | / | 40 | 80 | 60 | 80 | 80 | 20 | 80 | 60 | 40 | 20 |
| | 0.5 | 100 | 100 | 100 | 100 | 100 | / | / | / | / | / | 60 | 80 | 80 | 80 | 60 | 0 | 80 | 60 | 20 | 20 |
| | 0.75 | 100 | 100 | 100 | 100 | 100 | / | / | / | / | / | 80 | 80 | 80 | 100 | 80 | 20 | 80 | 60 | 40 | 40 |
| False | 0.25 | 100 | 100 | 100 | 100 | 100 | 100 | 100 | 100 | 100 | 100 | 43 | 85 | 96 | 85 | 85 | 0 | 64 | 85 | 0 | 0 |
| | 0.5 | 100 | 100 | 100 | 100 | 100 | 100 | 100 | 100 | 100 | 100 | 72 | 96 | 96 | 96 | 96 | 0 | 96 | 96 | 24 | 0 |
| | 0.75 | 100 | 100 | 100 | 100 | 100 | 100 | 100 | 100 | 100 | 100 | 32 | 64 | 48 | 64 | 64 | 0 | 48 | 48 | 16 | 0 |
| **TransOptAS** | | | | | | | | | | | | | | | | | | | | | |
| True | 0.25 | 100 | 100 | 100 | 100 | 100 | / | / | / | / | / | 40 | 80 | 100 | 60 | 60 | 0 | 60 | 60 | 60 | 20 |
| | 0.5 | 100 | 100 | 100 | 100 | 100 | / | / | / | / | / | 40 | 80 | 80 | 80 | 40 | 20 | 60 | 80 | 60 | 20 |
| | 0.75 | 100 | 100 | 100 | 100 | 100 | / | / | / | / | / | 40 | 80 | 80 | 80 | 40 | 20 | 40 | 80 | 40 | 20 |
| False | 0.25 | 20 | 100 | 100 | 100 | 100 | 100 | 100 | 100 | 100 | 100 | 21 | 64 | 100 | 43 | 64 | 0 | 21 | 85 | 0 | 0 |
| | 0.5 | 100 | 100 | 100 | 100 | 100 | 100 | 100 | 100 | 100 | 100 | 24 | 120 | 120 | 48 | 48 | 0 | 48 | 48 | 24 | 24 |
| | 0.75 | 100 | 100 | 100 | 100 | 100 | 100 | 100 | 100 | 100 | 100 | 48 | 64 | 80 | 64 | 64 | 16 | 64 | 48 | 48 | 16 |

Table 5 reports the percentage of function evaluations that can be omitted when self-supervised learning (SCARF, TabNet) outperforms the supervised RF baseline trained with 100% labeled data in the case of the PSO portfolio. Here we present the analysis per algorithm configuration, offering more fine-grained insights.

Table 5: Aggregated efficiency gains of self-supervised learning for PSO, reported as the percentage of function evaluations that can be omitted ($\Delta_{f,s,\ell,r}$) when SCARF or TabNet achieves lower MAE than the supervised RF baseline trained with 100% labeled ground performance data. Results are averaged across five folds and five repetitions for each configuration of feature representation, evaluation split (I, R, PC, P), and stratification strategy (RLS = FALSE: split-aligned, RLS = TRUE: random). Label ratios of 25%, 50%, and 75% are considered, while the 100% case is omitted since all instances must be labeled. Higher values indicate greater savings in ground-truth evaluations.

| Evaluation split | | | I | | | | | R | | | | | PC | | | | | P | | | |
|---|---|---|---|---|---|---|---|---|---|---|---|---|---|---|---|---|---|---|---|---|---|
| | | | | | | | | | DeepELA | | | | | | | | | | | | |
| RLS | % labeled | PSO1 | PSO2 | PSO3 | PSO4 | PSO5 | PSO1 | PSO2 | PSO3 | PSO4 | PSO5 | PSO1 | PSO2 | PSO3 | PSO4 | PSO5 | PSO1 | PSO2 | PSO3 | PSO4 | PSO5 |
| True | 0.25 | 0 | 0 | 0 | 0 | 0 | / | / | / | / | / | 40 | 80 | 60 | 80 | 60 | 60 | 20 | 60 | 40 | 40 |
| | 0.5 | 60 | 0 | 60 | 0 | 0 | / | / | / | / | / | 60 | 80 | 60 | 80 | 60 | 60 | 40 | 60 | 60 | 40 |
| | 0.75 | 100 | 40 | 100 | 40 | 20 | / | / | / | / | / | 80 | 80 | 60 | 80 | 100 | 60 | 60 | 60 | 60 | 40 |
| False | 0.25 | 0 | 0 | 0 | 0 | 0 | 0 | 0 | 0 | 0 | 0 | 43 | 64 | 43 | 64 | 64 | 43 | 0 | 64 | 21 | 43 |
| | 0.5 | 0 | 0 | 0 | 0 | 0 | 0 | 0 | 0 | 0 | 0 | 72 | 72 | 72 | 72 | 72 | 48 | 24 | 72 | 0 | 48 |
| | 0.75 | 100 | 60 | 80 | 60 | 0 | 40 | 0 | 60 | 0 | 0 | 48 | 80 | 48 | 80 | 64 | 48 | 48 | 32 | 48 | 32 |
| | | | | | | | | | Doe2Vec | | | | | | | | | | | | |
| True | 0.25 | 20 | 0 | 20 | 0 | 0 | / | / | / | / | / | 40 | 0 | 60 | 40 | 60 | 40 | 40 | 0 | 40 | 60 |
| | 0.5 | 40 | 20 | 60 | 40 | 0 | / | / | / | / | / | 60 | 40 | 60 | 20 | 60 | 40 | 40 | 20 | 40 | 40 |
| | 0.75 | 60 | 60 | 60 | 60 | 0 | / | / | / | / | / | 60 | 60 | 80 | 60 | 60 | 40 | 40 | 40 | 40 | 40 |
| False | 0.25 | 20 | 0 | 0 | 0 | 0 | 0 | 0 | 0 | 0 | 0 | 21 | 0 | 21 | 0 | 43 | 21 | 21 | 0 | 21 | 0 |
| | 0.5 | 20 | 0 | 20 | 0 | 0 | 0 | 0 | 0 | 0 | 0 | 24 | 0 | 24 | 24 | 48 | 48 | 24 | 0 | 48 | 24 |
| | 0.75 | 20 | 20 | 20 | 20 | 0 | 100 | 40 | 100 | 80 | 0 | 48 | 16 | 48 | 16 | 48 | 32 | 32 | 16 | 32 | 16 |
| | | | | | | | | | ELA | | | | | | | | | | | | |
| True | 0.25 | 0 | 0 | 0 | 0 | 0 | / | / | / | / | / | 40 | 40 | 40 | 40 | 60 | 40 | 20 | 40 | 20 | 0 |
| | 0.5 | 0 | 0 | 0 | 0 | 0 | / | / | / | / | / | 40 | 60 | 40 | 60 | 60 | 40 | 40 | 60 | 40 | 0 |
| | 0.75 | 0 | 0 | 0 | 0 | 0 | / | / | / | / | / | 40 | 40 | 40 | 60 | 60 | 60 | 40 | 40 | 60 | 0 |
| False | 0.25 | 0 | 0 | 0 | 0 | 0 | 0 | 0 | 0 | 0 | 0 | 43 | 43 | 43 | 43 | 64 | 43 | 43 | 43 | 43 | 0 |
| | 0.5 | 0 | 0 | 0 | 0 | 0 | 0 | 0 | 0 | 0 | 0 | 48 | 48 | 48 | 48 | 48 | 72 | 48 | 48 | 72 | 0 |
| | 0.75 | 0 | 0 | 0 | 0 | 0 | 0 | 0 | 0 | 0 | 0 | 32 | 32 | 32 | 48 | 48 | 32 | 48 | 32 | 48 | 0 |
| | | | | | | | | | TransOptAS | | | | | | | | | | | | |
| True | 0.25 | 60 | 0 | 80 | 20 | 0 | / | / | / | / | / | 80 | 80 | 80 | 80 | 60 | 20 | 60 | 60 | 60 | 60 |
| | 0.5 | 100 | 60 | 100 | 80 | 40 | / | / | / | / | / | 80 | 80 | 100 | 100 | 60 | 40 | 80 | 60 | 60 | 100 |
| | 0.75 | 100 | 60 | 100 | 80 | 80 | / | / | / | / | / | 80 | 80 | 80 | 80 | 80 | 20 | 80 | 80 | 80 | 40 |
| False | 0.25 | 20 | 0 | 40 | 0 | 0 | 0 | 0 | 0 | 0 | 0 | 85 | 21 | 100 | 64 | 21 | 21 | 21 | 64 | 21 | 0 |
| | 0.5 | 40 | 0 | 40 | 0 | 20 | 40 | 0 | 40 | 0 | 0 | 96 | 96 | 96 | 72 | 72 | 48 | 96 | 72 | 72 | 96 |
| | 0.75 | 60 | 20 | 80 | 20 | 40 | 100 | 60 | 100 | 100 | 0 | 64 | 48 | 64 | 64 | 48 | 16 | 48 | 48 | 32 | 80 |

