~~, additional optimizers such as CMA-ES, also include Bayesian optimization, active labeling, and explicit accounting of~~ and additional large-scale optimizers. We do not design or evaluate domain-specific SSL pretext tasks in this work, as our aim is to assess whether existing SSL approaches can reduce labeling

requirements for optimizer performance prediction in a manner that is comparable across landscape representations. Introducing custom, landscape-specific pretext tasks would reduce experimental control, make cross-representation comparisons less meaningful, and shift the focus away from the central question of label efficiency. This choice also ensures that any gains we observe arise from the SSL methods themselves rather than from domain-specific tailoring, thereby making our conclusions more general. We consider landscape-aware SSL tasks a promising direction for future work, with the potential to further improve representation quality and predictive performance. In addition, future work could explicitly quantify the wall-clock time and energy savings achieved through reduced labeling requirements and fewer function evaluations, providing a more comprehensive assessment of the practical efficiency benefits of SSL in this setting.

**LLM usage:** LLMs were used for grammar editing and minor coding support (e.g., debugging, snippets), while all . All scientific content, design, and final code were produced and validated by the authors, who take full responsibility.

**Reproducibility statement** All the code for executing the experiments presented in this paper can be found Complete code available at `https://anonymous.4open.science/r/optissl-iclr-F776`. Please refer to Implementation details presented in Appendix B for the implementation details and sensitivity study in Appendix C.

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

| 0.25 | 20 | 0 | 40 | 0 | 0 | 0 | 0 | 0 | 0 | 0 | 85 | 21 | 100 | 64 | 21 | 21 | 21 | 64 | 21 | 0 |
| | 0.5 | 40 | 0 | 40 | 0 | 20 | 40 | 0 | 40 | 0 | 0 | 96 | 96 | 96 | 72 | 72 | 48 | 96 | 72 | 72 | 96 |
| | 0.75 | 60 | 20 | 80 | 20 | 40 | 100 | 60 | 100 | 100 | 0 | 64 | 48 | 64 | 64 | 48 | 16 | 48 | 48 | 32 | 80 |