# OpenReview forum: "Self-supervised Learning to Predict Optimizer Performance: A Trade-off Study Between Ground Truth Requirements and Prediction Quality"
_ICLR.cc/2026/Conference — Submitted to ICLR 2026_

### Official Review · Reviewer_x9mQ · 2025-10-28

**Soundness:** 3
**Presentation:** 3
**Contribution:** 2
**Rating:** 6
**Confidence:** 1

**Summary:**

This paper addresses a significant and practical problem in the field of automated algorithm selection (AAS) for black-box optimization (BBO): the prohibitive cost of acquiring ground-truth performance labels. The authors conduct a comprehensive empirical study to investigate whether self-supervised learning (SSL) can mitigate this labeling burden.

The core idea is to decouple representation learning from performance prediction. An SSL model (SCARF or TabNet) is first pre-trained on problem landscape representations (ELA, DeepELA, DoE2Vec, TransOptAS) without performance labels. This pre-trained model is then fine-tuned on a small fraction (25%-75%) of the expensive ground-truth performance data (from DE and PSO portfolios). The goal is to quantify the trade-off between the number of labels used and the final prediction quality, especially under varying degrees of distribution shift (I, R, PC, P splits).

**Strengths:**

1. The paper tackles a highly practical issue. The cost of running extensive benchmarks (e.g., all DE/PSO variants on all BBOB problems) is a major bottleneck in meta-learning for AAS. The "green benchmarking" motivation is strong and timely.

2. The results are not ambiguous. The paper clearly demonstrates a "label-efficiency sweet spot," finding that SSL (especially SCARF + DoE2Vec) can match or exceed the fully-supervised RF baseline with only 50-75% of the labels.

3. The paper correctly identifies that the real value of SSL is not in easy in-distribution tasks (where RF+ELA is fine) but in the more realistic PC split (moderate shift). Here, SSL provides substantial gains, allowing for ~80% label omission (Table 2), which is a significant finding.

**Weaknesses:**

1. The paper honestly reports that the zero-shot (P split) scenario remains challenging for all methods. While SSL shows some minor gains, no method truly solves this hardest problem. This suggests that the current SSL pre-training tasks and representations still do not capture the "essence" of problem landscapes well enough to generalize to entirely unseen problem classes.

2. The study focuses on SCARF (contrastive) and TabNet (masking). While these are good representatives, the tabular SSL field is evolving. It's unclear if other methods (e.g., VIME, SubTab) might yield different results.

3. TabNet is consistently unstable at low label ratios (e.g., 25%) across many experiments (Fig 1). The paper notes this but doesn't fully explore why this specific SSL method fails so dramatically in the low-data regime compared to SCARF, which is generally stable.
4. The custom metric $\Delta_{f,s,l,r}$ is slightly convoluted. While the goal is to quantify "savings," the calculation (scaling by $\overline{p}_{f,s,l,r} \cdot (1-l)$) is indirect. A more direct comparison, such as "MAE of SSL at 50% labels vs. MAE of RF at 100% labels," might be clearer. However, Figure 1 and Table 1 (which are clear) support the conclusions.

**Questions:**

see weakness

---

> ### Author Response · Authors · 2025-11-27
>
> ### Q1: The paper honestly reports that the zero-shot (P split) scenario remains challenging for all methods. While SSL shows some minor gains, no method truly solves this hardest problem. This suggests that the current SSL pre-training tasks and representations still do not capture the "essence" of problem landscapes well enough to generalize to entirely unseen problem classes.
>
> Our goal was to systematically evaluate whether existing SSL methods can reduce labeling requirements for optimizer performance prediction. Incorporating new SSL mechanisms would have shifted the scope toward architectural or methodological innovation rather than assessing label efficiency. We agree that this extension is promising, and we explicitly highlight it as valuable direction for future work in the paper’s conclusion.
>
> ### Q2: The study focuses on SCARF (contrastive) and TabNet (masking). While these are good representatives, the tabular SSL field is evolving. It's unclear if other methods (e.g., VIME, SubTab) might yield different results.
>
> We evaluated two additional SSL methods, VIME and SAINT, to broaden the coverage of SSL paradigms in this setting. SAINT represents hybrid SSL that combines predictive and contrastive objectives, while VIME provides a second representative of predictive SSL. Across all evaluated settings, VIME and SAINT consistently achieve performance that lies between SCARF and TabNet. Including these additional methods strengthens the main conclusions of the paper, which remain unchanged.
>
> ### Q3: TabNet is consistently unstable at low label ratios (e.g., 25%) across many experiments (Fig 1). The paper notes this but doesn't fully explore why this specific SSL method fails so dramatically in the low-data regime compared to SCARF, which is generally stable.
>
> TabNet is indeed unstable at the 0.25 label ratio in the P split, which we attribute to its large default batch size. In our ablation study, reducing the batch size stabilizes TabNet in this low-label setting, as shown in Appendix C. However, even with this adjustment, TabNet remains inferior to the supervised baseline, and the overall conclusions of the paper remain unchanged.
>
> ### Q4: The custom metric $\Delta_{f,s,l,r}$ is slightly convoluted. While the goal is to quantify "savings," the calculation (scaling by $\overline{p}_{f,s,l,r} \cdot (1-l)$) is indirect. A more direct comparison, such as "MAE of SSL at 50% labels vs. MAE of RF at 100% labels," might be clearer. However, Figure 1 and Table 1 (which are clear) support the conclusions.
>
> We would like to clarify that the goal of the study is to provide an estimator for the number of saved function evaluations. This is the reason we proposed the specific metric. The metric already implicitly compares the MAE of each SSL method against the MAE of the 100% RF model, since this comparison is used to determine how many function evaluations can be omitted.

---

### Official Review · Reviewer_Uv9D · 2025-10-28

**Soundness:** 2
**Presentation:** 3
**Contribution:** 2
**Rating:** 6
**Confidence:** 3

**Summary:**

This paper studies label efficiency for predicting black-box optimizer performance via self-supervised learning (SSL) on problem-landscape representations. The authors conduct a well-controlled comparative study across four problem representations (ELA, DeepELA, DoE2Vec, TransOptAS), three learners (RF, SCARF, TabNet), and four train-test splits with increasing distributional shift (I, R, PC, P).

**Strengths:**

- It’s a well-posed, practically relevant question that discerning how many ground-truth evaluations (expensive) are really needed for reliable performance prediction under distribution shift.
- Extensive experimentations: 4 reps × 3 learners × 4 splits × 4 label ratios; multi-target regression over five configurations per portfolio; DE and PSO considered (PSO results in appendix).
- The author defines a clear Δ (omittable evaluations) metric and reports large savings in PC even at 25% labels.

**Weaknesses:**

- All problems are synthetic BBOB recombinations at fixed dimension (d=10) with a single evaluation budget. The affine blend construction also ties geometry/topology to BBOB priors. I recommend ablations over dimension, budget, and non-BBOB families, and at least one real-world surrogate (e.g., hyperparameter optimization on tabular models) to bound claims.
- Using RF alone (plus a naive baseline) undercuts the decisiveness of conclusions; RF is solid but not SOTA for tabular regression. Stronger baselines (XGBoost/LightGBM/CatBoost and TabR/RealMLP (I might miss some SOTA tab reg. model since I am not in that field)) could alter the crossover points where SSL “wins” and materially change Δ. At minimum, include XGBoost and LightGBM and recompute.
- Though the abstract concedes that P is challenging; Table 1 suggests TabNet can be brittle at low label ratios, with SSL sometimes failing to beat RF. The paper should analyze why: e.g., representation shift (pre-training corpora vs. P split topology), capacity-label mismatch, or optimization noise. Also, for Interpretability: which landscapes drive SSL gains? We see consistent PC improvements, but the mechanism remains opaque.

**Questions:**

1. How do conclusions change for higher $d$ (e.g., 20/40) ? Do learned reps or ELA degrade differently?
2. If resources are available, how do other SOTA model compare to RF under I/R/PC/P?
3. For the P split, have you tried domain augmentation (e.g., class-level perturbations) or adapter fine-tuning to stabilize SSL?

---

> ### Author Response · Authors · 2025-11-27
>
> ### Q1: How do conclusions change for higher $d$ (e.g., 20/40) ? Do learned reps or ELA degrade differently?
>
> We used the data from Cenikj et al., which represents the most comprehensive study on algorithm performance prediction, algorithm selection, and the evaluation of changes in train–test data distributions. Our study focuses on 10-dimensional single-objective continuous optimization problems. The 10d setting was selected because, within the evolutionary computation community, it is considered a higher-dimensional scenario and is sufficient for the type of analysis we perform. While we agree that much higher dimensions and large-scale optimization problems are possible, the algorithm portfolio we evaluate, along with the algorithms included in it, was not developed for large-scale settings, where their performance drops significantly. Moreover, TransOptAS, DoE2Vec, and DeepELA are predefined models that were originally trained on 10d problems. To extend them to higher dimensions, they would need to be pre-trained separately for each dimension, which in turn requires generating many new benchmark problem instances using the generator we employed. We leave this extension for future work and explicitly acknowledge this limitation in the discussion section of the paper.
>
> ### Q2: If resources are available, how do other SOTA model compare to RF under I/R/PC/P?
>
> We have also evaluated XGBoost and LightGBM as supervised baselines across all splits and representations. LightGBM performs similarly to RF, while RF remains the best overall, which is why we reported those results in the main text. XGBoost shows lower performance compared to the others. In the revised manuscript, we have added these results to the plots in Appendix D and provided additional explanation in the experimental design section.
>
> To complement SCARF and TabNet, we also evaluated two additional SSL methods, VIME and SAINT, to broaden the coverage of SSL paradigms in this setting. SAINT represents hybrid SSL that combines predictive and contrastive objectives, while VIME provides a second representative of predictive SSL. This enables us to assess whether TabNet’s weak performance is specific to its architecture or reflects a broader limitation of predictive SSL for this task. Across all evaluated settings, VIME and SAINT consistently achieve performance that lies between SCARF and TabNet. Including these additional methods strengthens the experimental evidence, while the main conclusions of the paper remain unchanged. The results are presented in Appendix D.
>
> ### Q3: For the P split, have you tried domain augmentation (e.g., class-level perturbations) or adapter fine-tuning to stabilize SSL?
>
> We did not apply domain-specific augmentations (such as class-level perturbations) or adapter-based fine-tuning in this work. This was an intentional design choice, as our goal was not to introduce new SSL mechanisms, but to systematically evaluate whether existing SSL methods can reduce labeling requirements for optimizer performance prediction. Incorporating domain-tailored augmentations or adapter modules would have shifted the scope toward architectural or methodological innovation rather than assessing label efficiency. We agree that these extensions are promising, and we explicitly highlight them as valuable directions for future work in the paper’s conclusion.

---

### Official Review · Reviewer_xyjn · 2025-11-02

**Soundness:** 2
**Presentation:** 2
**Contribution:** 2
**Rating:** 4
**Confidence:** 3

**Summary:**

This paper is an empirical study on using existing SSL methods to reduce labeling costs for the AAS task . It does not propose any new method or model. It just applies existing tools to this problem.
The evidence for SSL being effective is weak. The experiments show a small advantage only in the moderate.
All experiments are on low-dimensional d=10 problems, which makes the conclusions highly questionable for any practical use. Given the lack of novelty and major experimental limits, this work is not up to the ICLR standard.

**Strengths:**

The problem itself is practical (reducing label costs), which is a real bottleneck
The experimental setup is systematic comparing.
The discussion of these different shifts is useful.

**Weaknesses:**

The main weakness is that the paper is lack of novelty.
The paper just applies existing SSL methods. It does not propose any new method. This is more of a benchmark report.
Besides, the experiments are not sufficient.
All the experiments use d=10. The conclusions cannot be generalized to larger d, so the results are not very useful.
In addition, the analysis in the paper is superficial and lack of details. The paper says SCARF is better than TabNet but it does not explain why in the discussion .

**Questions:**

Why only use d=10? The results from such a low dimension are not very convincing. Could the authors show any results on higher dimensions?

---

> ### Author Response · Authors · 2025-11-27
>
> ### Q: Why only use d=10? The results from such a low dimension are not very convincing. Could the authors show any results on higher dimensions?
>
> We used the data from Cenikj et al., which represents the most comprehensive study on algorithm performance prediction, algorithm selection, and the evaluation of changes in train–test data distributions. Our study focuses on 10-dimensional single-objective continuous optimization problems. The 10d setting was selected because, within the evolutionary computation community, it is considered a higher-dimensional scenario and is sufficient for the type of analysis we perform. While we agree that much higher dimensions and large-scale optimization problems are possible, the algorithm portfolio we evaluate, along with the algorithms included in it, was not developed for large-scale settings, where their performance drops significantly. Moreover, TransOptAS, DoE2Vec, and DeepELA are predefined models that were originally trained on 10d problems. To extend them to higher dimensions, they would need to be pre-trained separately for each dimension, which in turn requires generating many new benchmark problem instances using the generator we employed. We leave this extension for future work and explicitly acknowledge this limitation in the discussion section of the paper.

---

### Official Review · Reviewer_moFr · 2025-11-09

**Soundness:** 3
**Presentation:** 1
**Contribution:** 2
**Rating:** 2
**Confidence:** 4

**Summary:**

This paper investigates whether self-supervised learning (SSL) can reduce the number of algorithm performance labels needed to predict optimizer performance in black-box optimization. The authors apply two SSL methods (SCARF and TabNet) to an existing dataset from Cenikj et al. (2025), evaluating on four landscape representations and four data splitting strategies. The main finding is that SSL methods, particularly SCARF on learned representations, can match supervised Random Forest baselines with 50-75% of labels under certain conditions.

**Strengths:**

- **Timely motivation:** The study addresses a real issue in black-box optimization; the high computational cost of acquiring performance labels—and positions the work within the broader sustainability (“green benchmarking”) agenda.
- **Comprehensive experiments:** The evaluation spans multiple representations, SSL methods, and both DE and PSO portfolios under four distinct data splits.
- **Reproducibility:** Code and implementation details (Appendix B) are thorough, and the repository is provided.
- **Consistent empirical results:** The main trend—that SCARF helps under moderate shift (PC)—is replicated across optimizers and representations.

**Weaknesses:**

#### 1. **Minimal Novelty and Heavy Dependence on Prior Infrastructure**
Nearly every component is reused from **Cenikj et al. (2025)**:
- identical dataset (8,280 affine recombinations);
- identical representations (ELA, DeepELA, DoE2Vec, TransOptAS);
- identical data splits (I, R, PC, P);
- identical portfolios (5 DE, 5 PSO configurations);
- identical metrics and protocols.

The sole addition is applying off-the-shelf SSL methods (SCARF [Bahri et al., 2022], TabNet [Arik & Pfister, 2021]) without modification or theoretical framing.
No new SSL pretext task, architectural adaptation, or analytical insight is proposed.
Thus, while the empirical study is well-executed, it constitutes an **incremental replication**, not a conceptual contribution suitable for ICLR.

*To strengthen the work*, the authors could:
- design domain-specific SSL objectives (e.g., perturbations of landscape features);
- analyze sample-efficiency or generalization bounds;
- visualize or interpret SSL embeddings to reveal *why* they help.

---

#### 2. **Unjustified and Contradictory Definition of “Distribution Shift”**
Section 4 claims that the four evaluation splits follow an *increasing difficulty order*
**I < R < PC < P**, implying that the *Problem Split (P)* scenario is the strictest zero-shot regime.
However, this claim is empirically and conceptually unsupported.

From **Table 1 (RLS = False)**:
| Split | Feature | Model | MAE (Mean ± SD) |
|:------|:---------|:------|:----------------|
| **PC** | DeepELA | Baseline | **0.261 ± 0.088** |
| **P**  | DeepELA | Baseline | **0.186 ± 0.014** |

Similar results appear for **DoE2Vec**, **ELA**, and **TransOptAS**, and across **RF** and **SCARF**.
Thus, the empirically measured errors *contradict* the claimed ordering—PC is consistently *harder* than P according to MAE, despite being defined as “easier.”
The authors never explain or reconcile this discrepancy.

More fundamentally, the notion of “distribution shift” in **meta-learning and algorithm selection** is itself **ill-defined**.
There is no universally accepted metric of shift between problem distributions, and the boundaries between in-distribution, partial-combination, and zero-shot settings are heuristic.
Given this ambiguity, it would be more accurate and scientifically sound to **avoid asserting any hardness ordering** among the four splits altogether.

Instead, the paper should:
- Present the splits simply as *distinct evaluation scenarios* rather than a progression of shift magnitude.
- Explicitly acknowledge that “distribution shift” in this context is a **loosely defined experimental proxy**, not a measurable quantity.
- Remove or rephrase claims suggesting that one split is inherently “harder” or “more out-of-distribution” than another.

This change would make the framing both more honest and better aligned with the current state of the meta-learning literature, where distribution shift remains an open and underspecified concept.

---

#### 3. **Superficial Analysis of Results**
Although Tables 1–2 clearly show SCARF’s gains under moderate shift (especially **PC** with **DeepELA** and **TransOptAS**), the paper provides no mechanistic explanation.
There is no ablation, no embedding visualization, and no investigation into what SSL representations capture beyond supervised baselines.
As a result, the findings remain *descriptive* rather than *insightful*.

---

#### 4. **Poor Presentation and Writing Quality**
- The prose is verbose, with long technical paragraphs (e.g., the affine recombination formula in §4) that obscure the core ideas.
- Many sentences are ungrammatical or awkwardly phrased (“We need to mention that…”).
- Figures and tables are densely formatted, lacking captions that interpret results.
- The paper reads more like an extended technical report than an ICLR submission, and would benefit from significant language editing and tighter structure.

**Questions:**

1. Why does the PC split (Table 1, DeepELA baseline = 0.261) appear empirically *harder* than the supposed zero-shot P split (0.186)?
2. What concrete real-world scenarios correspond to each of the four splits (I, R, PC, P)?
3. Were any domain-specific SSL pretext tasks attempted (e.g., landscape perturbation or masked-feature prediction)?
4. Can you visualize the learned embeddings (e.g., t-SNE/PCA) to show what SSL captures?
5. How sensitive are your results to SSL hyperparameters (corruption rate, batch size, etc.)?
6. Do the observed gains persist beyond the Cenikj et al. dataset, e.g., in higher-dimensional or noisy problems?

---

The paper delivers a careful empirical study but offers limited novelty and weak conceptual grounding. Its main insight, that generic SSL (SCARF, TabNet) can reduce labeling cost, is interesting yet unsurprising.

Substantial re-framing, clearer writing, and domain-specific methodological advances would be needed for acceptance at ICLR.

---

> ### Author Response · Authors · 2025-11-27
>
> ### Q1: Why does the PC split (Table 1, DeepELA baseline = 0.261) appear empirically harder than the supposed zero-shot P split (0.186)?
>
> We would like to clarify that the notion of “hardness” (i.e., difficulty level) of the splits was initially described somewhat subjectively, based on expectations about the types of problems appearing in the train and test sets. In the revised version, we have rephrased this to avoid implying a formal difficulty scale. It is important to emphasize that comparing baseline performance between the PC and P splits is not meaningful, as this involves different train datasets with different distributions of the target used for regression. The performance metrics are computed on different test sets, and therefore, the absolute numbers cannot be interpreted as indicating that one split is inherently harder than the other. These metrics are relative within each split. The PC and P splits represent different datasets, constructed based on different levels of overlap between problem combinations and their instances. For this reason, all result discussions in the paper are made within each split, not across splits. Because the training data also differ between splits, the baselines themselves change accordingly. Their robustness is evaluated across five random seeds, which is also reflected in the plots.
>
> ### Q2: What concrete real-world scenarios correspond to each of the four splits (I, R, PC, P)?
>
> We would also like to clarify that our evaluation is conducted in a controlled academic setting. The black-box optimization community still lacks standardized real-world benchmark suites targeted by evolutionary computation (based on our algorithm portfolio used in this paper), so academic benchmarks remain the only reliable way to analyze representation behavior before moving toward real-world applications. In single-objective continuous optimization, benchmark functions often include multiple instances created through shifts, rotations, or noise perturbations. The Instance Split (I) evaluates whether representations generalize to these familiar landscapes expressed in new instances, most in-domain scenarios. The Random Split (R) introduces more variability by assigning instances to training and testing without constraints. Here, test problems may be similar to or completely different from those seen during training, mimicking heterogeneous real-world task distributions. The Problem Combination Split (PC) becomes more challenging by excluding all training instances involving one parent function class and testing on combinations involving this unseen class. This setting reflects tasks built from a mix of known and partially unknown landscape components. The last setting is the Problem Split (P), where none of the parent classes used to generate the test problems appear in the training data. This strict zero-shot scenario tests whether representations can handle entirely new landscape types with no structural overlap. Together, these four splits form a progression from familiar landscape variations to completely novel problem classes, allowing us to systematically examine how representation learning methods generalize as the landscapes grow increasingly different from those observed during training.
>
> ### Q3: Were any domain-specific SSL pretext tasks attempted (e.g., landscape perturbation or masked-feature prediction)?
>
> We did not design or evaluate domain-specific SSL pretext tasks in this work, which was an intentional design choice. Our goal was to systematically evaluate whether existing SSL approaches (representing different SSL paradigms) can reduce labeling requirements for optimizer performance prediction in a way that is comparable across landscape representations. Introducing custom, landscape-specific pretext tasks would have made comparisons across representations less controlled and would have shifted the focus away from the central question of label efficiency. This choice also makes our findings more general: any label-efficiency improvements arise from the SSL itself rather than from landscape-specific tailoring. That said, we fully agree that landscape-aware SSL objectives are a promising direction that could further improve performance; thus, we explicitly include this topic in future work.

---

> ### Author Response · Authors · 2025-11-27
>
> ### Q4: Can you visualize the learned embeddings (e.g., t-SNE/PCA) to show what SSL captures?
>
> To assess what the SSL embeddings capture beyond the original benchmark representations, we performed a nearest-neighbor analysis comparing similarity in feature space with actual differences in optimizer performance. For each test instance, we located its most similar labeled training instance and computed their cosine similarity both in the original representations (ELA, DeepELA, DoE2Vec, TransOptAS) and in the SSL-derived spaces. We then measured the absolute difference in their true performance, providing a direct way to evaluate how well each representation aligns similarity with real performance behavior. The analysis was conducted in the in-distribution setting across all label ratios, using a single representative fold, a fixed labeled-instance seed, and no random splitting. Results are shown for the DE3 configuration. A well-aligned representation should place performance-similar problems close together, meaning high similarity values should correspond to small performance differences. For the 75% training ratio, the results show the same overall pattern: SCARF consistently improves or preserves similarity–performance alignment, TabNet collapses similarity and removes meaningful resolution, and the original representations remain competitive where they are already well structured. We added a part in the main text (Section 5, Impact of SSL on similarity semantics) and further extended it in Appendix E.
>
> ### Q5: How sensitive are your results to SSL hyperparameters (corruption rate, batch size, etc.)?
>
> We conducted hyperparameter ablation studies for SCARF and TabNet, evaluating their sensitivity across corruption rate, temperature, batch size (for SCARF), and architectural, regularization, and batch-size parameters (for TabNet). Experiments were performed on the DE portfolio using ELA and TransOptAS features under the R and PC splits, with stratified labeled-instance selection. MAE was averaged across all folds and seeds for each labeled ratio. Across all tested configurations, SCARF is highly stable, with only minimal performance variation, and the default hyperparameters lie well within the stable region. TabNet exhibits variability, particularly at low label ratios, but none of the ablations change the main conclusions of the paper. The results are presented in Appendix C.
>
> ### Q6: Do the observed gains persist beyond the Cenikj et al. dataset, e.g., in higher-dimensional or noisy problems?
> We used the data from Cenikj et al., which represents the most comprehensive study on algorithm performance prediction, algorithm selection, and the evaluation of changes in train–test data distributions. Our study focuses on 10-dimensional single-objective continuous optimization problems. The 10d setting was selected because, within the evolutionary computation community, it is considered a higher-dimensional scenario and is sufficient for the type of analysis we perform. While we agree that much higher dimensions and large-scale optimization problems are possible, the algorithm portfolio we evaluate, along with the algorithms included in it, was not developed for large-scale settings, where their performance drops significantly. Moreover, TransOptAS, DoE2Vec, and DeepELA are predefined models that were originally trained on 10d problems. To extend them to higher dimensions, they would need to be pre-trained separately for each dimension, which in turn requires generating many new benchmark problem instances using the generator we employed. We leave this extension for future work and explicitly acknowledge this limitation in the discussion section of the paper.

---

### Author Response · Authors · 2025-11-20
**General Response**

We thank the reviewers for their constructive feedback. We emphasize that our contribution is a controlled empirical study that examines whether existing SSL approaches can reduce labeling requirements for optimizer performance prediction in single-objective continuous optimization. Because each performance label requires multiple expensive optimization runs, evaluating label-efficient learning is highly relevant for the benchmarking community. To ensure fair comparisons, we intentionally avoid domain-specific SSL augmentations and focus solely on label efficiency across established representations and SSL paradigms, which is why we do not introduce a new self-supervised learning (SSL) method. The updated version of the paper, including all revisions described here, will be uploaded soon. We will also submit a separate, point-by-point response addressing each reviewer’s questions individually, ensuring full clarity and transparency.

# What is Added or Clarified in the Revision


**1. Clarified interpretation of the four splits (I/R/PC/P)**

We rewrote the descriptions to avoid suggesting a “difficulty scale”. Because each split uses a different training distribution and regression target, MAE values across splits are not comparable. All analyses are now explicitly within-split.

**2. Added analysis of SSL embedding behavior**

We added a new analysis comparing instance-pair similarities:

- in the original representations (ELA, DeepELA, DoE2Vec, TransOptAS),
- in SSL embeddings,
- and their correlation with performance differences.

This appears in the Appendix with an explanation in the discussion.


**3. Added SSL hyperparameter sensitivity studies**

We performed ablations on multiple hyperparameters for both SCARF and TabNet models. The results show that the conclusions from the study remain unchanged: SCARF is consistently stable, and while TabNet shows small variation, it still doesn’t perform better than the supervised baseline. Plots and commentary were added to the paper.

**4. Added additional supervised baselines**

We evaluated LightGBM and XGBoost. LightGBM performs similarly to RF; XGBoost is weaker. RF remains the most robust. Updated plots and supplementary tables were added.

**5. Added a third SSL paradigm**

We included SAINT [1] (a hybrid SSL approach, combining predictive and contrastive pretext tasks). SAINT performs better than TabNet but worse than SCARF.

**6. Clarified why no domain-specific pretext tasks were used**

We explain that landscape-specific augmentations would confound representation comparisons and divert attention from label efficiency, which is the focus of this work. Domain-aware SSL is now explicitly listed as future work, where the goal is to train an algorithm selector based on the regression models (i.e., obtaining the best predictive regression model).

**7. Clarified the ΔFEV metric**

We expanded the explanation showing how ΔFEV estimates saved function evaluations by comparing each SSL model to the fully supervised RF baseline.

**8. Clarified the focus on d = 10**

We added a detailed justification:
- All benchmark representation models (DeepELA, DoE2Vec, TransOptAS) are pretrained only for 10D
- The algorithm portfolio in Cenikj et al. is not designed for high dimensions
- Extending to 20–40D requires new benchmark generation and full re-training of multiple models

This is now acknowledged as a limitation and a direction for future work.


*[1] Somepalli, G., Goldblum, M., Schwarzschild, A., Bruss, C. B., & Goldstein, T. SAINT: Improved Neural Networks for Tabular Data via Row Attention and Contrastive Pre-Training. NeurIPS 2022 First Table Representation Workshop. 2022.*

---

### Comment · Area_Chair_d4Nz · 2025-11-27

Dear reviewers,

The authors have provided a joint response to your reviews, where they highlight the changes they have made. I would appreciate if you could let both me and the authors know how this response and these changes impact your assessment of the paper.

Best,

AC

---

> ### Author Response · Authors · 2025-11-27
>
> Dear AC and reviewers,
>
> We will submit the revised version and the point-by-point responses to each reviewer later this afternoon.

---

### Author Response · Authors · 2025-11-27
**Revised submission and point-by-point response to reviewers**

Dear AC and reviewers,

We sincerely thank the reviewers for their thoughtful and constructive feedback, which helped us improve the quality and clarity of the paper.

We have uploaded the revised version of the paper, along with a supplementary file highlighting the differences between the original and revised manuscripts. We have also provided detailed point-by-point responses addressing each reviewer’s questions (in official comments).

In summary, our revision:
1. Clarifies the interpretation of the four splits (we avoid implying an explicit difficulty scale) across the paper
2. Provides an analysis of the SSL embedding behavior (**Section 5**: _Impact of SSL on similarity semantics_ paragraph)
3. Explores how hyperparameters affect performance for both SCARF and TabNet by performing ablation studies (**Appendix C**), demonstrating that the conclusions of the main study do not change with different hyperparameters
4. Includes four additional models: two supervised (XGBoost and LightGBM) and two self-supervised (SAINT and VIME), expanding **Section 4** and providing more details in **Appendix D**
5. Elaborates more on the current limitations and future work, and offers more details about the rationale behind our design choices in **Section 6**

Best,

Authors

---

### Author Response · Authors · 2025-12-03
**General Summary**

Dear AC,

We greatly appreciate the time and effort invested by the Area Chair and all reviewers in evaluating our submission. Given the early end of the discussion period, we provide an additional detailed summarization of (1) key positive aspects acknowledged across reviews, and (2) major revisions and new analyses added to the updated manuscript.

### 1. Positive points highlighted by reviewers:
- A timely and practically relevant problem: multiple reviewers noted that label-efficiency in algorithm performance prediction is an important problem in black-box optimization, especially given the high computational cost of ground-truth evaluations.
    - `moFr`: "Timely motivation ... positions the work within the broader sustainability (“green benchmarking”) agenda."
    - `Uv9D`: "It’s a well-posed, practically relevant question ..."
    - `xyjn`: “The problem itself is practical (reducing label costs), which is a real bottleneck”
    - `x9mQ`: "The paper tackles a highly practical issue ... motivation is strong and timely."
- Extensive and systematic experimental design: reviewers ( `moFr`, `Uv9D`) acknowledged the breadth of the experiments (4 landscape representations, 4 data splits, 4 label ratios, 5 DE and 5 PSO configurations, multiple SSL/SL learners).
- Reproducibility and transparency
    - `moFr`: “Code and implementation details (Appendix B) are thorough, and the repository is provided.”
- Clear empirical insights: reviewers highlighted consistent findings: SCARF improves label efficiency under moderate distribution shift; RF remains a strong baseline; Zero-shot remains a challenging scenario.
    - `moFr`: "The main trend ... is replicated across optimizers and representations."
    - `xyjn`: "The discussion of these different shifts is useful."
    - `Uv9D`: "The author defines a clear Δ ... large savings in PC even at 25% labels."
    - `x9mQ`: "The results are not ambiguous. The paper clearly demonstrates a label-efficiency sweet spot ...", "The paper correctly identifies that the real value of SSL is not in easy in-distribution tasks ... SSL provides substantial gains, allowing for ~80% label omission (Table 2), which is a significant finding."

### 2. Summary of revisions:
- Clarifications (throughout the paper):
    - Reframed the interpretation of the four train-test splits, without explicit difficulty labeling (as suggested by `moFr`).
    - Clarified that MAE values across splits are not directly comparable.
    - Clarified the study's focus on label efficiency, not SSL innovation.
- New analyses and experiments:
    - SSL embedding behavior analysis (added in Section 5 + Appendix E), examining the correlation between performance differences and cosine similarity (in both original and SSL embedding spaces); addressing questions (`moFr`) and comments (`xyjn`, `Uv9D`).
    - Hyperparameter ablations for SCARF and TabNet (Appendix C): study of multiple hyperparameters, confirming that the main conclusions do not change with different hyperparameters (raised by `moFr`).
    - Expanded model suite: 2 supervised (XGBoost, LightGBM) and 2 SSL (VIME, SAINT) models added (main text + Appendix D), addressing concerns from `Uv9D` and `x9mQ`. Findings: XGBoost generally performs worse, and LightGBM ≈ RF. SAINT and VIME fall between SCARF and TabNet.
- Expanded the discussion of limitations and future work in Section 6 (limitation to 10d, domain-specific SSL pretext tasks, future extensions), elaborating on concerns raised by multiple reviewers.

We have received valuable feedback from 4 reviewers, and we have carefully addressed each question, considerably improving the clarity, completeness, depth, and transparency of the paper. We have uploaded a revised manuscript, a supplementary file with highlighted changes, and point-by-point responses addressing every question (as official comments to the reviewers).

Best,

Authors

---

### Meta-Review · Area_Chair_Xwz2 · 2026-01-07

**Summary:**

The paper investigated the use of Self-Supervised Learning (SSL) to improve label efficiency in optimizer performance prediction. While the authors strengthened the empirical study with additional baselines and a preliminary similarity analysis, numerous concerns remain unaddressed, including the limited novelty of applying off-the-shelf SSL methods to existing infrastructure without domain-specific adaptations. The provided analysis is largely diagnostic rather than causal, failing to explain the underlying mechanisms of SSL gains or the instability of certain models (e.g., TabNet). Furthermore, critical questions regarding generalization to high-dimensional problems were met with scope-based defenses rather than concrete experimental evidence.

**Reviewer Concerns:**

### [moFr]

Weakness 1: The rebuttal only partially addresses the reviewer’s concern but does not fundamentally remove it. The authors essentially accept the heavy reuse of Cenikj et al. (2025)’s infrastructure and argue this is intentional to enable a controlled study focused on label efficiency, rather than proposing a new SSL method. They do strengthen the paper along one of the reviewer’s suggested directions by adding an embedding-behavior / similarity-semantics analysis to provide some intuition for why SSL helps, and they broaden robustness via more baselines and hyperparameter ablations. However, they do not introduce domain-specific SSL objectives, architectural adaptations, or any theoretical/sample-efficiency analysis. The reviewer’s concern is primarily about ICLR-level novelty beyond an incremental empirical application of off-the-shelf SSL, the rebuttal mainly justifies the scope rather than fully resolving the concern.

Weakness 3: Not fully addressed. The authors add an “embedding behavior / similarity semantics” analysis that compares pairwise cosine similarities in the original feature space versus the SSL embedding space and relates these to performance differences. However, the new analysis remains largely diagnostic rather than causal. There is still limited interpretability (e.g., what specific landscape properties are being captured) and no targeted ablations that isolate which aspects of the SSL pipeline drive the gains.

Q3: Not fully addressed. See "Weakness 1."

Other concerns are addressed.

### [xyjn]

Novelty: Not fully addressed. See comment above.

superficial analysis: Not fully addressed. See comment above.

Higher dimensional experiments: Not fully addressed. The authors provide a technical justification for omitting high-dimensional experiments. However, it does not address the reviewer’s request (“show any results on higher dimensions”), so the generalization concern remains.

### [Uv9D]

W1 & Q1: Not fully addressed. See comment above.

W2 & Q2 are addressed.

W3 & Q3: Not addressed. The rebuttal serves mainly as a scope defense rather than a substantive response to the critique. The reviewer asked for concrete analysis of why TabNet is brittle at low label ratios, why SSL sometimes fails to beat RF, and which landscape types drive the consistent PC gains, as well as whether stabilizing approaches for P (e.g., domain augmentations or adapters) were tried. The authors simply state these were intentionally not explored to avoid methodological innovation and defer them to future work, without adding diagnostics or evidence. As a result, the response explains why no change was made, but it does not meaningfully address the underlying “mechanism and interpretability are missing” concern.

### [x9mQ]

Q2 and Q3 are addressed.

Q1 and Q4 remain more defensive (or justificatory) than explanatory, and would benefit from lightweight analyses and clearer interpretability rather than scope statements.

**Reviewer Scores:**

The AC anticipates the outcomes:

Reviewer moFr: Concerns are partially addressed, so the score may rise slightly from 2 to 4.

Reviewer xyjn: Concerns are mostly not addressed, so the score will likely remain at 4 or drop to 2.

Reviewers Uv9D and x9mQ: Concerns are partially addressed, so the score is expected to hold at 6 given the context.

---

### Decision · Program_Chairs · 2026-01-26

Reject